# Tamoxifen for the treatment of myeloproliferative neoplasms: A Phase II clinical trial and exploratory analysis

Current therapies for myeloproliferative neoplasms (MPNs) improve symptoms but have limited effect on tumor size. In preclinical studies, tamoxifen restored normal apoptosis in mutated hematopoietic stem/progenitor cells (HSPCs). TAMARIN Phase-II, multicenter, single-arm clinical trial assessed tamoxifen's safety and activity in patients with stable MPNs, no prior thrombotic events and mutated $JAK2^{V617F}$, $CALR^{ins5}$ or $CALR^{del52}$ peripheral blood allele burden ≥20% (EudraCT 2015-005497-38). 38 patients were recruited over 112w and 32 completed 24w-treatment. The study's A'herns success criteria were met as the primary outcome ( ≥ 50% reduction in mutant allele burden at 24w) was observed in 3/38 patients. Secondary outcomes included ≥25% reduction at 24w (5/38), ≥50% reduction at 12w (0/38), thrombotic events (2/38), toxicities, hematological response, proportion of patients in each IWG-MRT response category and ELN response criteria. As exploratory outcomes, baseline analysis of HSPC transcriptome segregates responders and non-responders, suggesting a predictive signature. In responder HSPCs, longitudinal analysis shows high baseline expression of JAK-STAT signaling and oxidative phosphorylation genes, which are downregulated by tamoxifen. We further demonstrate in preclinical studies that in JAK2V617F+ cells, 4-hydroxytamoxifen inhibits mitochondrial complex-I, activates integrated stress response and decreases pathogenic JAK2-signaling. These results warrant further investigation of tamoxifen in MPN, with careful consideration of thrombotic risk.

Myeloproliferative neoplasms (MPN) arise from mutations acquired by HSPCs, most frequently affecting the genes encoding the kinase JAK2[1–4] or the multi-functional protein CALR[5,6]. Currently JAK1/2 inhibitors can improve disease-related symptoms and overall survival but have a limited impact on clone size[7,8], likely because they cannot discriminate between mutant and wild-type JAK2 or due to the acquisition of pharmacological resistance[9–11]. Allogeneic HSC transplantation remains the only curative treatment for MPN but can only be performed in a minority of patients due to its toxicity[12], warranting investigation of new therapies.

Men exhibit a higher prevalence of myeloid neoplasia compared with women[13,14]. Furthermore, MPN subtypes with poorer prognosis (primary myelofibrosis and polycythemia vera, compared with essential thrombocythemia) have a higher prevalence in males than in females[15–17]. Additionally, the risk of secondary myelofibrosis, which worsens the outcomes of PV/ET, is higher for men than for women, regardless of their age[17–19]. However, the reasons underlying this gender difference are unclear. It is possible that sex-chromosome genes and gender-dependent differences in epigenetic regulation, metabolism or immune response partly account for sexual dimorphism in

✉e-mail: Claire.Harrison@gstt.nhs.uk; sm2116@cam.ac.uk

cancer[20]. Another explanation might be the loss of sex chromosomes with age, which preferentially occurs in males, perhaps suggesting a higher genomic instability in men[21].

However, one key determinant of gender disparities in cancer might be the effect of sex hormones[20]. Estrogens regulate the self-renewal, proliferation, and apoptosis of mouse hematopoietic stem and progenitor cells (HSPCs)[22,23]. Estrogen receptors (ERs) are differentially expressed in mouse HSPC subsets[22]. ERα activation induces proliferation of mouse long-term HSCs[22,23] and protects them from proteotoxic stress through the modulation of UPR[24]. The selective ER modulator (SERM) tamoxifen induces apoptosis of multipotent hematopoietic progenitors but spares normal HSCs[22]. In MPN mouse models, tamoxifen restores the physiological apoptosis levels in mutant HSCs and selectively eliminates these cells, but not their non-mutated counterparts[22]. Based on these preclinical studies, we conducted a Phase II, multicenter, single-arm A'herns design clinical trial assessing tamoxifen's safety and activity in reducing molecular markers of disease burden in MPN (TAMARIN). Here we report the results of the TAMARIN study. In addition, we describe an exploratory analysis of HSPCs from study patients and associated laboratory research investigating the mechanism of action of tamoxifen in human MPN.

## Results

### Efficacy and safety of tamoxifen in patients with MPN

A total of 38 patients (27 males and 11 females) were recruited over 118 weeks. The Trial Scheme for Eligibility and Central Analysis is summarized in Supp. Fig. 1A. The study protocol is provided as a Supplementary Note in the Supplementary Information. Baseline characteristics of Study subjects are indicated in Table 1. The disease types comprised 37% essential thrombocythemia (ET), 29% polycythemia vera (PV), 16% primary myelofibrosis (PMF), 13% post-PV

myelofibrosis (MF) and 5% post-ET MF. The primary outcome of the study was ≥50% reduction of mutant allele burden in peripheral blood at 24w. Secondary outcomes included mutant allele burden reductions between 25% and 50% reduction at 24w, ≥50% reduction at 12w, thrombotic events, toxicities, hematological response, proportion of patients in each IWG-MRT response category, ELN response criteria, and improvement in response category, which are reported in Supp. Data 1. One patient did not start treatment, 35 completed 12w of treatment, 32 completed 24w of treatment and 31 continued on treatment after 24w. One patient discontinued following a thrombotic event and 11 discontinued due to toxicity. Four serious adverse effects were reported: 1 unrelated Grade 3 urinary tract infection, 1 unrelated Grade 1 intracranial hemorrhage, and 2 vascular disorders, both potentially related to tamoxifen treatment (1 superficial thrombophlebitis and 1 thromboembolic event). 2 patients withdrew from the trial and no deaths were reported. The overall symptoms responses were 19% complete response, 71.4% partial response, and 9.5% no response in ET/PV (Supp. Data 1).

Three patients achieved the primary outcome (≥50% reduction at 24w) and met the required number for trial success: one *JAK2*[V617F+] ET male and two *JAK2*[V617F+] PV males. Five additional patients exhibited a ≥25% reduction at 24w and comprised: one *JAK2*[V617F+] ET female, one *JAK2*[V617F+] PMF female, one *JAK2*[V617F+] PV male, one *CALR*[delS2] ET male and one *CALR*[ins5] ET female (Fig. 1). One patient showed a ≥50% reduction at 12w but this was not sustained at 24w (Supp. Data 1). A total of 6 patients remained on trial treatment beyond 48w as they were considered to be deriving clinical benefit.

### Exploratory analysis of HSPCs from study patients

An exploratory analysis was undertaken to investigate the differences between responders and non-responders. Changes in mutant allele burden were paralleled by the in vitro response of HSPCs from study patients to the soluble metabolite of tamoxifen (4-hydroxytamoxifen, 4OH-TAM). Colony-forming units in culture (CFU-Cs) from the available baseline peripheral blood samples were reduced after 24 h treatment with 4OH-TAM in tamoxifen-responders only (Fig. 2a–d), suggesting a direct effect of tamoxifen derivatives on human HSPCs, resembling previous results in animal models[22]. CFU-C genotyping showed that the balance between mutant and WT colonies is reduced 5 times by 4OH-TAM treatment in responders, compared with non-responders (Fig. 2e), suggesting a predominant effect on mutant HSPCs from study responders.

Bioinformatic analysis of the transcriptome of CD34+ HSPCs isolated from the peripheral blood of study patients revealed a perfect clustering of responders and non-responders at baseline (Fig. 2f and Supp. Fig. 1B), suggesting a potential predictive signature of response. Gene-set enrichment analysis (GSEA) and integrative pathway enrichment analysis showed a distinct molecular signature of HSPCs from responders and non-responders at baseline. The top gene sets enriched in responders were related to mitochondrial activity, protein turnover, and unfolded protein response (UPR) (Fig. 2g and Supp. Data 2). In contrast, GSEA in non-responders showed enrichment in DNA replication, chromosome separation, and chromatin condensation-associated pathways (Fig. 2h and Supp. Data 2), which suggests a comparatively higher cell replication and reduced genome accessibility in HSPCs from non-responders. GSEA at baseline showed a higher activation of HSPCs in responders, manifested as increased expression of STAT3-, STAT5-, and inflammation-related genes (Fig. 2i–k). Additionally, HSPCs from responders showed increased expression of apoptosis-related genes (Fig. 2l).

### Tamoxifen modulates the pro-apoptotic integrated stress response in JAK2[V617F+] human cell lines

Cellular stress, such as nutrient deprivation, unfolded protein response (UPR), redox imbalance, or mitochondrial stress, trigger a

## Table 1 | Patients' characteristics

| Variable | Level | N (%) |
|---|---|---|
| Age (yr) | Mean | 66.3 |
| | Range | 50.0–87.0 |
| Sex | Male | 27 (71.1) |
| | Female | 11 (28.9) |
| Disease subtype | Essential thrombocythaemia | 14 (36.8) |
| | Primary Myelofibrosis | 6 (15.8) |
| | Post-essential thrombocythaemia myelofibrosis | 2 (5.3) |
| | Polycythaemia vera | 11 (28.9) |
| | Post-polycythaemia vera myelofibrosis | 5 (13.2) |
| Time since diagnosis (years) | Mean | 7.1 |
| | Range | 0.4–22.1 |
| Number of therapies received prior to tamoxifen | No details given | 3 (7.9) |
| | 1 | 30 (78.9) |
| | 2 | 2 (5.3) |
| | 3 | 1 (2.6) |
| | 4 | 2 (5.3) |
| Treatments received | No details given | 3 (7.9) |
| | Ruxolitinib | 3 (7.9) |
| | Hydroxycarbamide | 27 (71.1) |
| | Interferon; hydroxycarbamide | 2 (5.3) |
| | Anagrelide; fedratinib; ruxolitinib | 1 (2.6) |
| | Hydroxycarbamide; interferon; anagrelide; ruxolitinib | 1 (2.6) |
| | Interferon; hydroxycarbamide; pacritinib; ruxolitinib | 1 (2.6) |

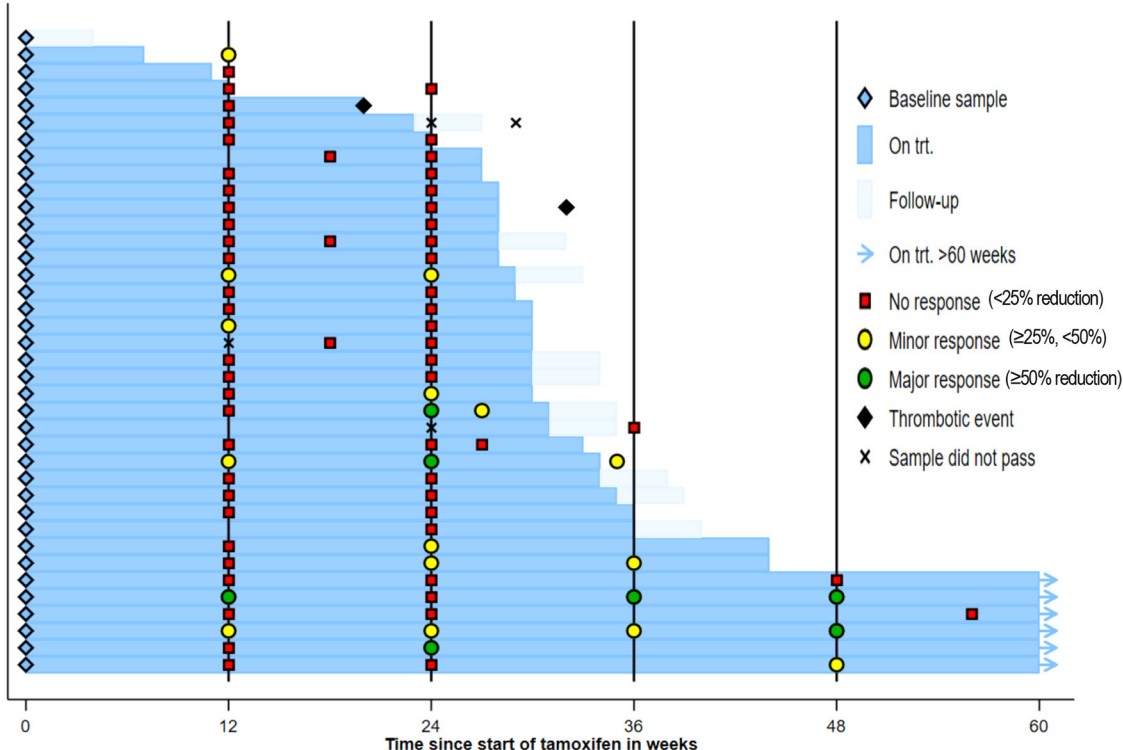

**Fig. 1 | Tamoxifen reduces mutant allele burden in a subset of MPN patients.** Swimmer plot of treatment, thrombotic events and allele burden response.

self-adaptive program named integrated stress response (ISR), which is characterized by the phosphorylation of the alpha subunit of the eukaryotic translation initiation factor (eIF2α) on Ser51, leading to globally attenuated cytosolic translation and preferential translation of activating transcription factor 4 (ATF4)[25,26]. The physiological activation of ATF4 axis is essential for HSC maintenance and adaptation[27], but promotes apoptosis after persistent stress-induced activation[28]. The eIF2α-ATF4 axis –the central regulatory hub of ISR– is regulated by four kinases: heme-regulated translational inhibitor (HRI/eIF2αK1), protein kinase RNA-activated (PKR/eIF2αK2), protein kinase R-like endoplasmic reticulum (ER) kinase (PERK/eIF2αK3) and general control nonderepressible 2 (GCN2/eIF2αK4)[29]. One key difference in HSPCs from responders was the increased expression of genes related to ISR, such as endoplasmic reticulum UPR, amino acid starvation, and the target genes of ATF3 and ATF4 (Fig. 3a–d). One of the top upregulated genes in responder HSPCs is the tumor suppressor *BTG1* (Supp. Fig. 1B), which promotes the function of the ATF4 in response to cellular stress[30].

To evaluate the possible direct proapoptotic effect of tamoxifen in human MPN, we treated different human cell lines carrying the *JAK2^{V617F}* mutation with 4OH-TAM, which dose-dependently reduced the survival of HEL human erythroleukemia cell line[31] (Fig. 3e). A similar dose-dependent reduction of viability was observed in the SET-2 cell line derived from essential thrombocythemia transformed into acute myeloid leukemia[32] (Fig. 3f). In contrast, a different *JAK2^{V617F}*-mutated human cell line derived from essential thrombocythemia transformed into acute myeloid leukemia (UKE-1) appeared resistant to 4OH-TAM-induced cell death (Fig. 3g), resembling the heterogeneous response of MPN patients to tamoxifen treatment. While basal UKE-1 cell survival was unaffected, serum deprivation sensitized UKE-1 cells to tamoxifen-induced cell death (Fig. 3h). Particularly, UKE-1 cells cultured in presence or absence of horse serum provided models of *JAK2^{V617F}*-mutated cells with the same genetic background but different sensitivity to tamoxifen.

As GSEA suggested that ISR is a hallmark of responsiveness to tamoxifen, HEL and UKE1 cells were transduced with lentivirus carrying ATF4 sensors to monitor the possible modulation of ISR by 4OH-TAM. As expected, thapsigargin –an endoplasmic reticulum stressor that activates ISR– simultaneously increased ATF4 translation in HEL cells and horse serum-starved UKE-1 cells; 4OH-TAM gradually increased ATF4 translation in tamoxifen-sensitive cell lines (Fig. 3i, j). Furthermore, 4OH-TAM dose-dependently induced eIF2α phosphorylation and ATF4 translation in starved UKE1 cells only (but not in UKE1 cells cultured in competent medium; Fig. 3k, l), mimicking sensitive HEL cells (Fig. 3m, n), and likely causing ISR. Therefore, we treated HEL cells with 4OH-TAM alone or in combination with the small molecule ISRIB, which reverts the effects of eIF2α phosphorylation[33]. ISRIB partially rescued 4OH-TAM-induced cell death (Fig. 3o), suggesting that tamoxifen can regulate ISR in MPN and cause mutant cell death.

**Tamoxifen targets mitochondrial respiratory complex I to induce apoptosis**

Estrogens can regulate cellular metabolism through different receptors, most prominently ERα[34], which transduces the effects of estrogens and tamoxifen on mouse HSPCs[22–24]. Consistent with preclinical results, in human *JAK2^{V617F+}* cell lines tamoxifen treatment reduced the expression of genes positively regulated by estradiol and ERα, and increased the expression of genes negatively regulated by estradiol and ERα (Supp. Fig. 2A–C and Supp. Data 3). However, full-length ERα mRNA (encoded by *ESR1*) and ERα protein expression were undetectable in these cells (Supp. Fig. 2D, E). Furthermore, the ERα degrader fulvestrant caused dose-dependent cell death in control MCF-7 human breast cancer cells, but it did not affect the survival of human *JAK2^{V617F+}* cell lines (Supp. Fig. 2F–I). These results suggest that ERα signaling cannot explain cell survival regulation by tamoxifen in human MPN.

To investigate the mechanisms underlying sensitivity of *JAK2^{V617F}*-mutated cells to tamoxifen-induced cell death, HEL, and UKE1 cells

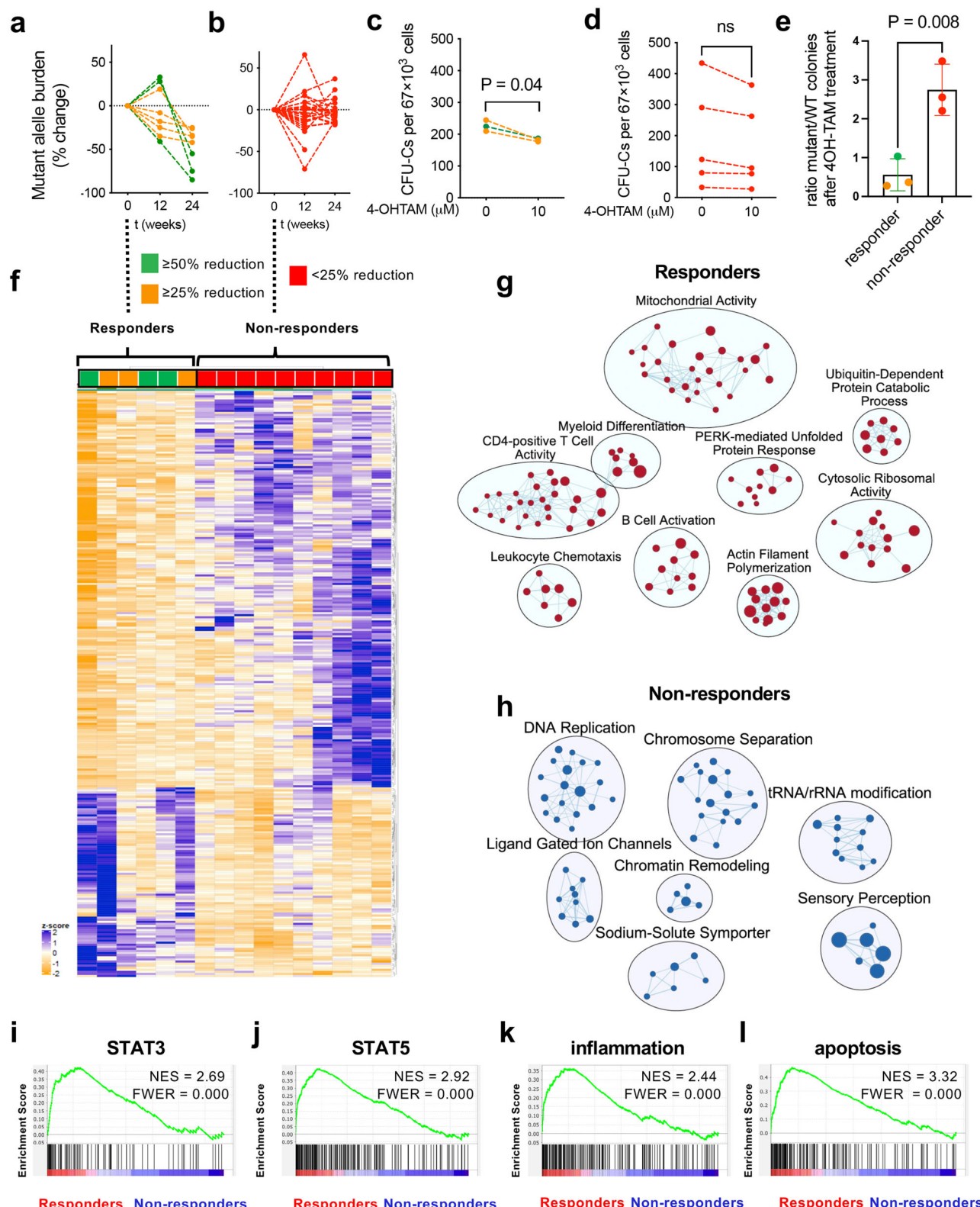

were treated with 4OH-TAM for 4 h and were analyzed by RNAseq (Supp. Fig. 1C, D). GSEA of genes differentiated expressed in HEL and starved UKE-1 cells showed marked downregulation of genes related to oxidative phosphorylation (OXPHOS) in tamoxifen-sensitive cells (Fig. 4a, b and Supp. Data 3). To validate these results in primary human samples and to interrogate their relevance to tamoxifen-induced reduction of mutant allele burden in MPN, a longitudinal analysis of the transcriptome of CD34+ HSPCs was conducted on 16 study subjects before tamoxifen treatment and after 24-week tamoxifen treatment. Resembling the results with $JAK2^{V617F+}$ human cell lines, the expression of OXPHOS-related genes was high at baseline in responders, and was specifically reduced by tamoxifen in responders, compared with non-responders (Fig. 4c, d and Supp. Data 4). These results validate the use of $JAK2^{V617F+}$ cell lines to investigate tamoxifen-induced cell death and suggest that tamoxifen-induced human $JAK2^{V617F+}$ cell death is associated with altered mitochondrial respiration.

**Fig. 2 | Distinctive transcriptomic signature of HSPCs from tamoxifen responders at baseline. a, b** Mutant allele burden change (%) in (**a**) responders achieving after 24w tamoxifen treatment allele burden reductions of ≥50% ($n = 3$, green) or ≥25%, <50% ($n = 5$, orange), and (**b**) in non-responders ($n = 28$, red) before treatment, and 12w or 24w after tamoxifen administration. **c-d** HSPCs measured as colony-forming units in culture (CFU-Cs) from study patients' peripheral blood mononuclear cells treated with 4OH-TAM (10 mM) or vehicle for 24 h. Reduced (**c**) or unchanged (**d**) HSPC numbers upon ex vivo 4OH-TAM treatment of baseline samples are consistent with the allele burden reductions observed in the same patients after 24w tamoxifen treatment (**c**, green ≥ 50%, $n = 2$; orange≥25%, $n = 1$; **d**, red < 25%, $n = 5$). *$p < 0.05$, two-tailed unpaired $t$ test. **e** CFU-C genotyping shows

a reduced balance of JAK2[V617F+] colonies, compared with WT colonies, by 4OH-TAM treatment in responders', but not in non-responders' samples (allele burden reductions after 24w are marked with green ≥ 50%, $n = 1$; orange ≥ 25%, $n = 2$; red < 25%, $n = 3$). **$p < 0.01$, two-tailed paired $t$ test. Data are mean ± SEM. **f** Heatmap reveals the disparity of gene expression between responders and non-responders at baseline. **g, h** Integrated pathway enrichment map using gene-sets enriched in responders (**g**) and non-responders (**h**). **i, l** Gene set enrichment analysis (GSEA) shows a higher activation of HSPCs in responders, with increased expression of (**i**) STAT3-, (**j**) STAT5-, (**k**) inflammation- and (**l**) apoptosis-related genes. NES, normalized enrichment score. FWER, family-wise error rate.

---

To directly test the effects of tamoxifen on mutant cell OXPHOS, the oxygen consumption rate was measured in HEL cells, SET-2 cells, and UKE-1 cells treated with 4OH-TAM or vehicle and subjected to the Cell Mito Stress kit (Agilent Seahorse). In HEL cells and SET-2 cells, 4OH-TAM caused a dose-dependent inhibition of the basal, maximal, and spare respiratory capacities and their associated ATP generation (Fig. 4e–h). In agreement with the starvation-dependent sensitivity of UKE-1 cells, the inhibition of mitochondrial respiration by 4OH-TAM was more pronounced, and OXPHOS-derived ATP was only decreased, in starved UKE-1 cells, compared with UKE-1 cells cultured with competent medium (Fig. 4i–l).

For confirmation, we used the Seahorse XF Real-Time ATP Rate Assay kit (Agilent) to simultaneously measure ATP generation from OXPHOS and glycolysis. OXPHOS yielded doubled ATP (compared with glycolysis) in tamoxifen-sensitive UKE-1 cells, and OXPHOS-derived (but not glycolysis-derived) ATP was significantly reduced in tamoxifen-sensitive SET-2 cells, HEL cells and starved UKE-1 cells, but not in tamoxifen-resistant UKE-1 cells (Fig. 4m–p). These results suggest that tamoxifen-induced JAK2[V617F+] cell death is associated with the inhibition of mitochondrial respiration.

The heterogenous response of human JAK2[V617F+] cell lines (see Fig. 4) resembled the metabolic effects of 4OH-TAM on peripheral blood mononuclear cells from study patients. 4OH-TAM inhibited mitochondrial respiration more pronouncedly in the peripheral blood mononuclear cells available from study responders at baseline, compared with non-responders (Fig. 5a–d).

To directly assess the metabolic effects of tamoxifen on HSPC-enriched cells at a similar dose as that used in the TAMARIN study, MPN mouse models were treated for 2 weeks with 10-fold-different tamoxifen doses, both of which have therapeutic effects in MPN mice[22]. Tamoxifen dose-dependently inhibited mitochondrial respiration in HSPC-enriched cells, even at a dose comparable to that used in study patients (Fig. 5e, f). These results suggest that the tamoxifen dose used in TAMARIN or for breast cancer prevention regimens can inhibit mitochondrial respiration in HSPCs.

To examine tamoxifen's subcellular distribution in JAK2[V617F+] cells, we took advantage of a fluorescent tamoxifen derivative (FLTX1) that retains SERM properties[35,36]. In HEL cells, super-resolution Airyscan2 confocal imaging showed FLTX1 signal colocalizing with the mitochondrial marker TOM20 (Fig. 6a), suggesting a potential enrichment of tamoxifen inside mitochondria. Previous studies in rat liver cells have suggested that tamoxifen can bind to and inhibit mitochondrial complex I, reducing the mitochondrial respiration rate[37]. To investigate potentially direct mitochondrial effects of 4OH-TAM, OXPHOS was measured in HEL cells permeabilized with the XF Plasma Membrane Permeabilizer (Agilent), which circumvents the need for mitochondrial isolation. In permeabilized JAK2[V617F+] cells, 4OH-TAM reduced OXPHOS at the level of complex I, but did not compromise complex II activity induced by succinate/ADP (Fig. 6b). Treatment of complex I purified from bovine heart mitochondrial membranes revealed a dose-dependent inhibition of NADH:O2 oxidoreduction rate by 4OH-TAM with IC$_{50}$ 3.3 µM; these effects were preserved (IC$_{50}$ 4.3 µM) after addition of antimycin to inhibit complex III and an alternative oxidase

to regenerate the ubiquinone, confirming the specific inhibition of complex I by 4OH-TAM (Fig. 6C). These effects were complemented by the reduced expression of complex I-related genes in HPSCs form study responders after tamoxifen treatment (Supp. Fig. 3A).

To directly evaluate the subcellular distribution of tamoxifen, 4OH-TAM concentration was measured in isolated mitochondrial or cytosolic fractions from treated MPN cell lines (Supp. Fig. 3B) using liquid chromatography tandem mass spectrometry (LC-MS/MS). 4OH-TAM concentration was 3- to 20-fold-higher in the mitochondria than in the cytosolic fraction of MPN cell lines (Fig. 6d), indicative of the mitochondrial accumulation of tamoxifen derivatives. Interestingly, mitochondrial 4OH-TAM concentration was 4-fold higher in sensitive (serum-deprived), compared with resistant (grown in competent medium) UKE-1 cells (Fig. 6e), possibly explaining their different response.

These results were reminiscent of MCF-7 breast cancer cells, where the inhibition of mitochondrial respiration by 4OH-TAM cannot be reverted with complex I substrates, but can be compensated with the complex II substrate succinate[38]. Therefore, we investigated the effects of cell-permeable succinate prodrugs, which can bypass the inhibition of complex I and restore the electron transport from complex II[39] (Supp. Fig. 3C, D). Monomethyl succinate did not affect the baseline survival of HEL cells or SET-2 cells; in contrast, pretreatment of these cells with monomethyl succinate resulted in a dose-dependent protection from 4OH-TAM-induced cell death (Fig. 6f, g). This protection was explained by increased basal, maximal, and spare respiratory capacities and restored ATP generation from OXPHOS (Fig. 6h, i). These results suggest that tamoxifen inhibits mitochondrial complex I and electron transport, decreasing OXPHOS-derived energy production in JAK2[V617F+] cells. Additionally, increased ATF4 translation by 4OH-TAM in HEL cells or serum-deprived UKE-1 cells was prevented by pretreatment with MMS or the antioxidant N-Acetyl Cysteine (NAC) (Supp. Fig. 3E, F), suggesting that 4OH-TAM-induced mitochondrial stress contributes to proapoptotic ISR in MPN.

Interestingly, treatment with canonical inhibitors of mitochondrial complex I had lesser effect compared with 4OH-TAM; rotenone compromised viability of HEL cells and UKE-1 cells (but not SET2 cells), while Piericidin A induced cell death in UKE-1 cells only (Supp. Fig. 4), suggesting a different molecular target of 4OH-TAM, compared with canonical complex I inhibitors.

Two MPN patients carrying CALR mutations showed a partial response to tamoxifen administration at 24w, indicative of the potential effect of tamoxifen on HSPCs carrying CALR mutations. To directly examine the effects of 4OH-TAM on CALR-mutated human cell metabolism and survival, we took advantage of the MARIMO cell line[40], which harbors a 61-basepair (bp) deletion in CALR exon 9, similar to the 52-bp deletion generated by the CALR[del52] mutation[41]. Resembling the effects observed in human JAK2[V617F+] cell lines, 4OH-TAM suppressed cellular respiration in MARIMO cells and reduced OXPHOS-derived ATP synthesis (Supp. Fig. 5A–D). The supplementation of monomethyl succinate protected MARIMO cells from 4OH-TAM-induced cell death (Supp. Fig. 5E), suggesting that tamoxifen induces apoptosis of cells carrying CALR mutations through a mechanism

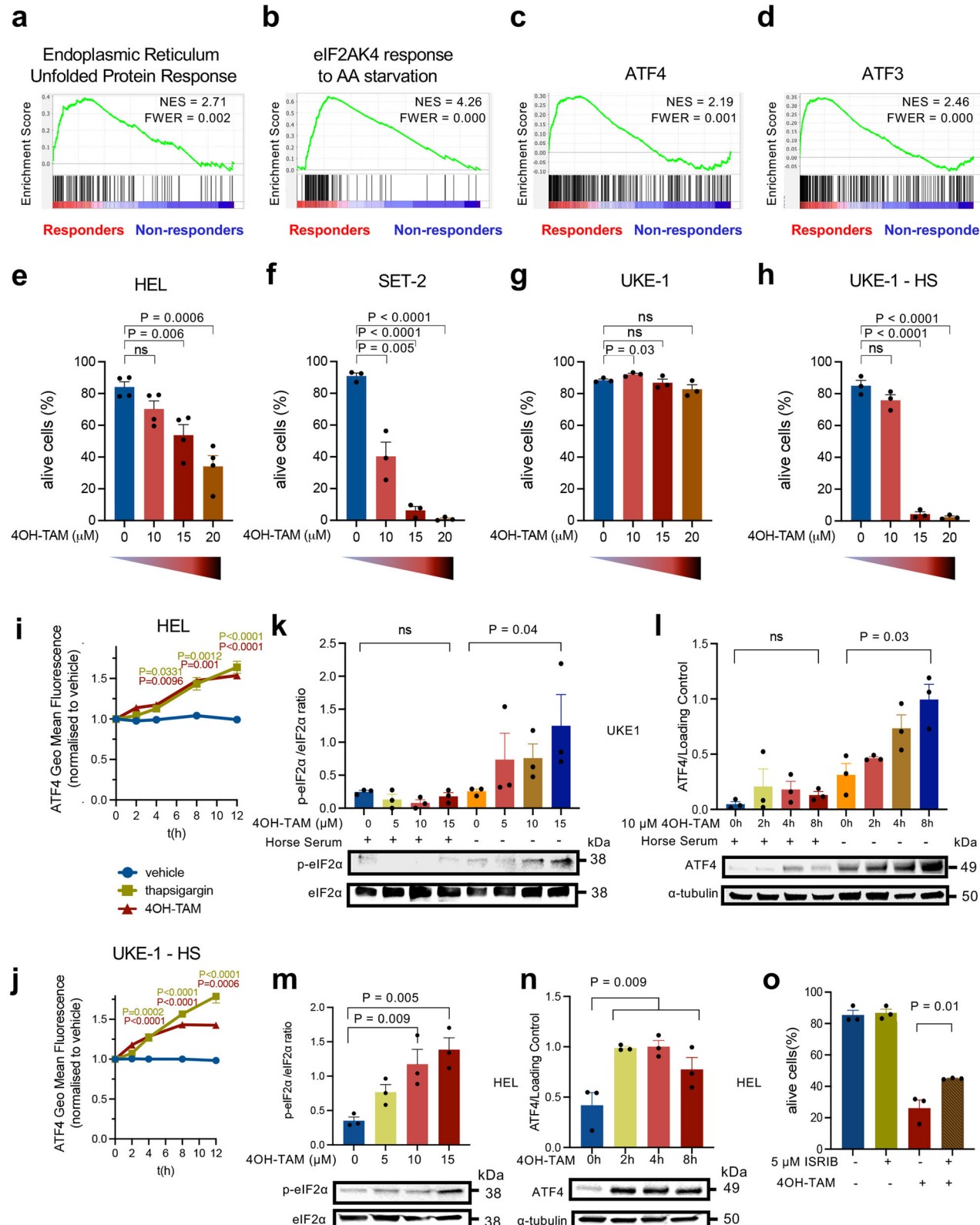

shared with JAK2[V617F] mutation, which involves respiratory complex I inhibition.

### Tamoxifen reduces OXPHOS-derived ATP and pathogenic JAK2-STAT5

The JAK2[V617F] mutation and activation of its downstream target STAT5 protect mutant HSPCs from physiological apoptosis[42,43]. However,

tamoxifen can restore normal apoptosis levels in JAK2[V617F]-mutated cells in a mouse MPN model[22]. In study responders, HSPC transcriptome was enriched in STAT5 target genes, compared with non-responders (see Fig. 2j). Furthermore, responders achieving ≥50% allele burden reduction displayed a higher enrichment score in STAT5 geneset, compared with patients showing ≥25% but <50% reduction (Fig. 7a). Additionally, 24-week tamoxifen treatment was associated

**Fig. 3 | JAK2$^{V617F+}$ human cell lines reproduce the differential sensitivity to tamoxifen of study patients. a–d** Gene set enrichment analysis (GSEA) from HSPCs from responders at baseline shows signs of integrated stress response, manifested as increased expression of genes involved in unfolded protein response (UPR) (**a**), response to aminoacid (AA) starvation (**b**), or target genes of activating transcription factors (ATF) 3 and 4 (**c, d**). NES, normalized enrichment score. FWER, family-wise error rate. **e–h** 24 h treatment with the soluble tamoxifen derivative 4-hydroxytamoxifen (4OH-TAM) dose-dependently reduces the viability of HEL cells (**e**), SET2 cells (**f**) and horse-serum-starved UKE-1 cells (**h**), but spares UKE-1 cells (**g**) cultured with competent medium (n = 4 independent experiments in HEL cells, n = 3 independent experiments in SET2 and UKE-1 cells). **i, j** 4OH-TAM induces ATF4 translation in HEL cells (**i**) and horse serum-deprived UKE-1 cells (**j**) expressing ATF4-mScarlet. The fluorescence intensity of 4OH-TAM and thapsigargin-treated cells was normalized to the vehicle control at each time point (n = 5 independent experiments). **k, l** Quantification (top) and representative Western blots (bottom) of (**k**) eIF2α and phosphorylated (p) eIF2α after 8 h vehicle/4OH-TAM treatment or (**l**) ATF4 and β-tubulin (loading control) before/after 4OH-TAM treatment in UKE-1 cells cultured with/without horse serum (n = 3 independent experiments). **m, n** Quantification (top) and representative western blots (bottom) of (**m**) eIF2α and p-eIF2α after 8 h vehicle/4OH-TAM treatment, or (**n**) ATF4 and β-tubulin (loading control) before/after 4OH-TAM treatment in HEL cells (n = 3 independent experiments). **o** Integrated stress response (ISR) inhibitor (ISRIB) partially rescues 4OH-TAM-induced HEL cell death. Frequency of viable cells after 24 h treatment with 4OH-TAM and ISRIB alone or in combination (n = 3 independent experiments). **e–o** Data are mean ± SEM. *p < 0.05, **p < 0.01, ***p < 0.001, ****p < 0.0001; One-way ANOVA and Dunnett's test.

with the downregulation of JAK2-related genes in study responders (Fig. 7b).

To directly compare the effects of tamoxifen on WT JAK2 and mutant JAK2$^{V617F}$, Ba/F3 cells expressing the JAK2-signaling receptor for erythropoietin (EpoR)[44] were transduced with wild-type *JAK2* or mutant *JAK2$^{V617F}$* [45]. 4OH-TAM-induced cell death was more pronounced in Ba/F3 cells carrying the human mutant JAK2$^{V617F}$ (Fig. 7c), supporting an increased activity of tamoxifen on the mutant cells. To investigate the impact of tamoxifen on pathogenic JAK2-STAT5 signaling protecting mutant cells from apoptosis[42,43], EpoR-expressing Ba/F3 cells were treated with 4OH-TAM or vehicle and were subsequently stimulated with erythropoietin to activate EpoR-JAK2-STAT5 signaling. Constitutive STAT5 activation, measured as phosphorylation at residue Y694, was detected in Ba/F3 cells expressing mutant JAK2$^{V617F}$ without erythropoietin stimulation. STAT5 phosphorylation was increased in cytokine-stimulated Ba/F3 cells expressing both WT and mutant JAK2; this effect was significantly blocked by 4OH-TAM in the *JAK2$^{V617F+}$* cells (Fig. 7d–e). To independently confirm these results in human MPN cell lines and using a different cytokine to activate JAK2 signaling, HEL cells, and SET-2 cells were treated with 4OH-TAM or vehicle and were subsequently stimulated with thrombopoietin, to drive MPL-JAK2-STAT5 signaling. STAT5-Y694 phosphorylation was inhibited by 4OH-TAM in a dose-dependent fashion (Fig. 7f, g).

We asked whether the mitochondrial inhibition by 4OH-TAM in human JAK2$^{V617F}$ mutant cells was responsible for decreased pathogenic JAK2-STAT5 signaling. Supporting this possibility, a previous study has demonstrated that ATP binding to the pseudokinase domain of JAK2 is critical for pathogenic activation of JAK2$^{V617F}$, but is not required for wild-type JAK2 activity[46]. To test this potential dependency from mitochondrial-generated ATP, thrombopoietin-stimulated STAT5-Y694 phosphorylation was measured after treatment with 4OH-TAM or vehicle, alone or in combination with monomethyl succinate (to activate mitochondrial complex II and bypass complex I inhibition; see Supp. Fig. 3C). Restoration of the mitochondrial electron transport and ATP generation rescued pathogenic JAK2$^{V617F}$-STAT5 activation in 4OH-TAM-treated HEL cells and SET2 cells (Fig. 8a, b). These results suggest that mitochondrial complex I inhibition by tamoxifen is responsible for decreased pathogenic JAK2$^{V617F}$ activation.

To confirm this ATP dependency in the inhibition of pathogenic JAK2$^{V617F}$-STAT5 signaling by 4OH-TAM, thrombopoietin-stimulated STAT5-Y694 phosphorylation was measured after treatment with 4OH-TAM or vehicle, alone or in combination with cell-permeable ATP (8-Bromoadenosine 5′-triphosphate, 8-Br-ATP). Similar to complex II activation, ATP supplementation rescued pathogenic JAK2$^{V617F}$-STAT5 activation in 4OH-TAM-treated HEL cells (Fig. 8c, d). The suppression of thrombopoietin-stimulated STAT5-Y694 phosphorylation was also confirmed by immunoblotting (Fig. 8e). Lastly, we tested whether 4OH-TAM could affect the phosphorylation level of JAK2 at Tyr 221 and Tyr1007/8, which are required for the maximal kinase activity of mutant JAK2$^{V617F}$ [47]. 4OH-TAM suppressed the cytokine-induced kinase autophosphorylation at Tyr 221 and

Tyr1007/8 in HEL cells, which was reversed by complex-II activation with succinate (Fig. 8f, g). Altogether, these results suggest that tamoxifen's inhibition of mitochondrial complex I can compromise pathogenic JAK2$^{V617F}$ activation (Fig. 9).

## Discussion

This Phase-II study demonstrates the safety and activity of tamoxifen in reducing the mutant allele burden in a subset of MPN patients who could potentially be prospectively identified based on their HSPC transcriptomic signature before the treatment. This response signature is characterized by increased JAK-STAT signaling, ISR and mitochondrial respiration. The perfect segregation of the HSPC transcriptome from responders and non-responders at baseline could serve in the future as a platform for the stratification of patients based on their likelihood to respond to tamoxifen and for the identification of predictive biomarkers of response, if prospectively validated. Unexpectedly, tamoxifen derivatives accumulate in the mitochondria, where they inhibit complex I, reducing ATP production and pathogenic STAT5 phosphorylation, eliminating human JAK2$^{V617F}$-mutant cells. These results suggest that the metabolic effects of SERMs in cancer might be underappreciated and propose ways to modulate pathogenic JAK-STAT signaling through metabolic rewiring. These results warrant further investigation of tamoxifen as potential therapeutic for MPN in larger studies, after careful consideration of the risk of thrombosis.

With the exception of interferon alpha[48], most agents currently used in MPN have a limited effect on clone size. For this reason, a 50% reduction in *JAK2$^{V617F}$* or *CALR* mutant allele burden was chosen as the primary endpoint for the TAMARIN study. The sample size was statistically calculated to confidently test the main endpoint in the minimal number of patients necessary to demonstrate safety and activity. The A'herns success criteria of this study were met with the primary outcome (≥50% reduction in mutant allele burden at 24w) having been observed in 3 patients. The ≥25% reduction in mutant allele burden at 24w observed in 5 additional patients further suggests that tamoxifen can reduce molecular markers of disease in a subset of MPN patients, although the conclusions rely on a small patient cohort and warrant confirmation in future larger studies.

Based on the increased risk of thromboembolic events in MPN and in tamoxifen-treated breast cancer patients[49], MPN patients with previous thrombotic events were excluded from this study. Overall, tamoxifen administration was found to be safe and non-toxic, with the exception of 2 thrombotic events, including a deep vein thrombosis, possibly related to the treatment. This should be regarded as a note of caution when weighing the possible benefit of reducing the clone size against the risk of thrombosis. Thrombotic events only occurred in non-responders; therefore, the selection of prospective responders based on the predictive signature might help reduce the risk of thrombosis in future studies. Additionally, the development of SERMs with a lower thrombotic risk might provide an alternative exploratory route.

A previous study has highlighted the high energy demands of *JAK2*-mutant cells and identified possible metabolic vulnerabilities in

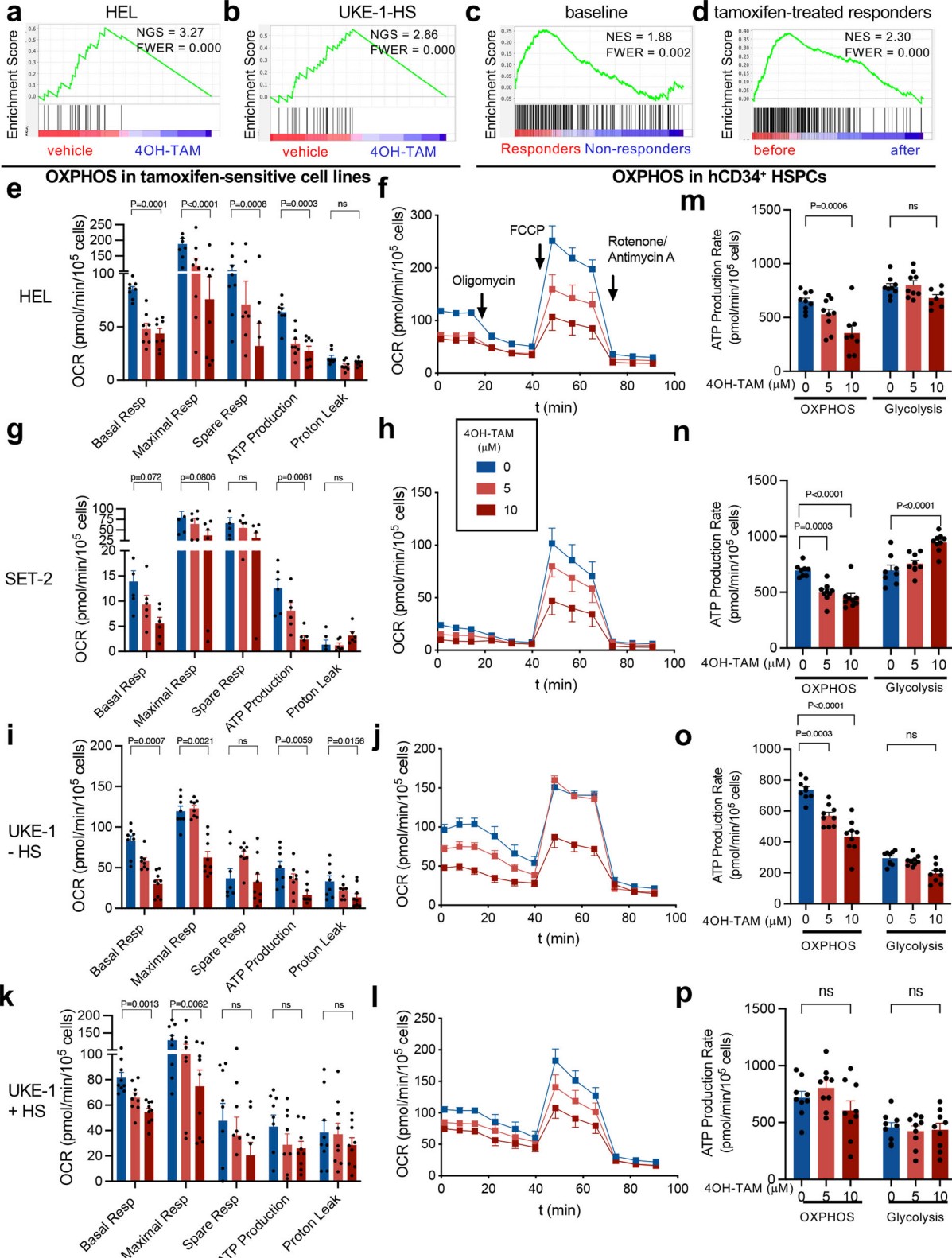

MPN[50]. Longitudinal transcriptomic analysis of HSPCs from study subjects showed high baseline expression of OXPHOS-related genes and their specific reduction after 24-week tamoxifen treatment in patients meeting the primary or secondary endpoints, compared with non-responders. These transcriptomic results were reproduced in tamoxifen-resistant and –sensitive $JAK2^{V617F+}$ human cell lines, where 4OH-TAM directly inhibited mitochondrial complex I and decreased

ATP synthesis and pathogenic JAK2 signaling. Indeed, the strong inhibition of respiratory complex I and electron transport chain by tamoxifen was demonstrated in isolated mitochondrion, as well as in permeabilized or intact cells. These results highlight the potential of SERMs to target metabolic vulnerabilities in MPN. Consistent with our results, 17β-estradiol and hydroxyestrogens have been shown to trigger mitochondrial stress and influence redox homeostasis[51].

**Fig. 4 | Tamoxifen downregulates OXPHOS-related genes and reduces OXPHOS-derived ATP generation in *JAK2*[V617F]-mutated human cells. a–d** High enrichment of OXPHOS-related genes in sensitive cells and its marked downregulation by tamoxifen treatment. Gene set enrichment analysis (GSEA) of differentially expressed genes after 4OH-TAM treatment of sensitive cell lines (**a**, **b**, *n* = 3 independent experiments), HSPCs from prospective TAMARIN responders achieving after 24w tamoxifen treatment allele burden reductions of ≥50% (**c**, *n* = 3 individuals) or ≥25%, <50% (*n* = 3 individuals), and non-responders (*n* = 10 individuals) or genes selectively downregulated by tamoxifen treatment in responder HSPCs (**d**, *n* = 6 individuals). NES, normalized enrichment score. FWER, family-wise error rate. **e–l** Dose-dependent inhibition of mitochondrial respiration and its derived ATP by 4OH-TAM in HEL cells (**e–f**, *n* = 8 biologically independent samples), SET-2 cells (**g**, **h**, *n* = 6 independent experiments) and UKE-1 cells cultured under serum starvation (**i**, **j**, *n* = 8 independent experiments) or (**k**, **l**, *n* = 9 independent experiments) with competent medium. Oxygen Consumption Rate (OCR) was measured after 2 h treatment with 4OH-TAM or vehicle, using the Cell Mito Stress kit (Agilent). Note the more severe reduction of respiratory capacity and mitochondrial ATP production in sensitive *JAK2*[V617F]-mutant cells (**e–j**), compared with resistant cells (**k**, **l**). **f, h, j, l** Timeline of OCR measurement after oligomycin to inhibit ATP synthase, FCCP to disrupt the mitochondrial membrane potential and rotenone/Antimycin A to inhibit mitochondrial complex I. **m-p** Selective inhibition of OXPHOS-derived (not glycolysis-derived) ATP generation by 4OH-TAM in sensitive *JAK2*[V617F]-mutant cells (*n* = 9 independent experiments). ATP generated from OXPHOS and glycolysis was simultaneously measured in sensitive (**m-o**) or resistant (**p**) cells using the ATP Real-Time rate assay kit (Agilent). **e–p** Data are mean + SEM. *$p < 0.05$, **$p < 0.01$, ***$p < 0.001$, Two-way ANOVA and Dunnett's test.

Furthermore, these results might inform autoimmune diseases dependent on JAK-STAT signaling (e.g. rheumatoid arthritis or psoriasis) or other malignancies, such as breast cancer, where the potential direct metabolic effects of SERMs might be underestimated. Genistein, an estrogen-like soy isoflavone, was reported to inhibit the JAK2-STAT5 axis independently of canonical estrogen signaling in lactating mammary epithelial cells[52]. It is worth mentioning the critical role of mitochondrial STAT3 for OXPHOS in RAS-driven transformed tumors[53]. Future studies will determine whether SERMs can suppress pathogenic JAK2-STAT3 activity by inhibiting mitochondrial STAT3.

ISR in HSPCs of study responders is strongly correlated with tamoxifen sensitivity. Furthermore, we show that the induction of mitochondrial stress and ISR by tamoxifen contributes to the proapoptotic mechanism in JAK2[V617F] cells. These results are consistent with the regulation of HSC survival by ATF4[27] and suggest that ISR modulation by tamoxifen might dictate whether mutant HSCs are able to adapt their metabolism, or instead undergo apoptosis. Consistent with an intrinsic adaptation of JAK2[V617F]-mutated cells to external stress in our study, enhanced proteotoxic burden and ISR signal have been previously described in HEL and SET-2 cells[54]. Of note, mitochondrial dysfunction can persistently enhance eIF2α phosphorylation and ATF4 translation, overloading the adaption capacity and leading to cell death[55]. Therefore, complex I inhibition by tamoxifen likely overcomes the self-adaption threshold of ISR and restores normal apoptosis in JAK2[V617F+] HPSCs. Indeed, a selective activation of eIF2α-ATF4 axis was observed in tamoxifen-sensitive HEL and starved UKE-1 cells. Similarly, a previous study has shown that inhibiting mitochondrial protein synthesis can overcome venetoclax resistance in AML through the induction of ISR[56].

Unexpectedly, a high subcellular enrichment of 4OH-TAM was detected inside the mitochondria of MPN cell lines, leading to respiratory complex I inhibition. Intriguingly, 4-fold higher 4OH-TAM concentration was detected in tamoxifen-sensitive (serum-deprived) UKE-1 cells, compared with tamoxifen-resistant UKE-1 cells (cultured with competent medium), suggesting that mitochondrial loading could critically determine sensitivity to tamoxifen. Given that the mitochondrial matrix is negatively charged, the mitochondrial localization of tamoxifen may be due to its localized positive charge in protonated tertiary amine of tamoxifen at physiological pH (Ionization Constant (pKa) of tamoxifen is 9.7)[57]. In addition to electrostatic attraction, the hydrophobic triphenylethylene backbone of tamoxifen facilitates the accumulation of tamoxifen in biomembranes, especially in the mitochondrial ones[58]. Of note, the strong partitioning of tamoxifen in lipid bilayer underlies the accumulation of tamoxifen and its active metabolites, reaching 2556 ng/g concentration in breast tumor[59]. A similar tamoxifen concentration (2474 ng/g) was found in the bone marrow of tamoxifen-treated human breast cancer[60]. This evidence adds to our results suggesting that 4OH-TAM mitochondrial loading might cause respiratory complex I inhibition in cancer cells.

It is interesting to note that the proapoptotic effect of 4OH-TAM on JAK2[V617F]-mutated cells was more pronounced compared with

canonical inhibitors of mitochondrial complex I, suggesting a different molecular target or complementary pathways (such as ISR modulation) contributing to overall response. More broadly, these results might be related to the higher prevalence of myeloid neoplasia in males, compared with females[13,14], possibly involving the regulation of cancer stem cells by sex hormones[20]. In the mouse hematopoietic system, estrogens regulate the self-renewal, proliferation, and apoptosis of HSPCs[22,23], where ERα activation induces proliferation of long-term HSCs[22,23] and protects them from proteotoxic stress through the modulation of the unfolded protein response[24]. The results of the TAMARIN study validate many of these findings in humans. However, a difference with the mouse model appears to be the transcriptional regulation by ERα. While the transcriptomic analysis is indicative of 4OH-TAM-induced transcriptional regulation by nuclear receptors in both human and mouse systems, ERα does not seem to mediate this regulation in humans since the full-length receptor is not expressed in tamoxifen-sensitive human *JAK2*[V617F]-mutated cell lines. However, ERα targets were transcriptionally regulated by tamoxifen in these cells, suggesting the participation of other nuclear receptors or coreceptors, which will be the subject of future studies.

Of note, all 3 patients achieving ≥50% reduction and 3 further patients achieving ≥25% reduction in the mutant allele burden at 24w carried the JAK2[V617F] mutation. Two other patients achieving ≥25% reduction at 24w carried del52 and Ins5 CALR mutations, but these mutations also require increased MPL-JAK-STAT activation to be pathogenic[61]. These results suggest that tamoxifen might interfere with hyperactive JAK-STAT signaling, which is enriched in the HSPC transcriptome from responders. Indeed, our study shows that tamoxifen might interfere with pathogenic JAK-STAT signaling by inhibiting mitochondrial complex I. This is supported by 4OH-TAM inhibition of mitochondrial complex I-derived ATP, reduced STAT5 phosphorylation, and their rescue by complex II activation or ATP supplementation, explaining the effects on cell survival.

In summary, the A'herns success criteria of this Phase-II study were met with the primary outcome having been observed in ≥3 patients. Tamoxifen responders could potentially be prospectively identified based on their HSPC transcriptome at baseline. Mutant HSPC sensitivity to tamoxifen is characterized by high activity of ISR pathway and high expression of JAK-STAT signaling and oxidative phosphorylation genes, which are downregulated by tamoxifen. Through transcriptional regulation and previously unrecognized direct metabolic effects, tamoxifen can selectively target cells with pathogenic JAK-STAT activation by modulating ISR and inhibiting mitochondrial complex-I and ATP generation. These results add to previous identification of other metabolic vulnerabilities in MPN[50] and advocate for future clinical studies to test the effects of SERMs in MPN, with careful consideration of thrombotic risk. Additionally, they might inform approaches in other diseases with pathogenic JAK-STAT signaling and in other malignancies where the direct metabolic effects of SERMs might be underestimated.

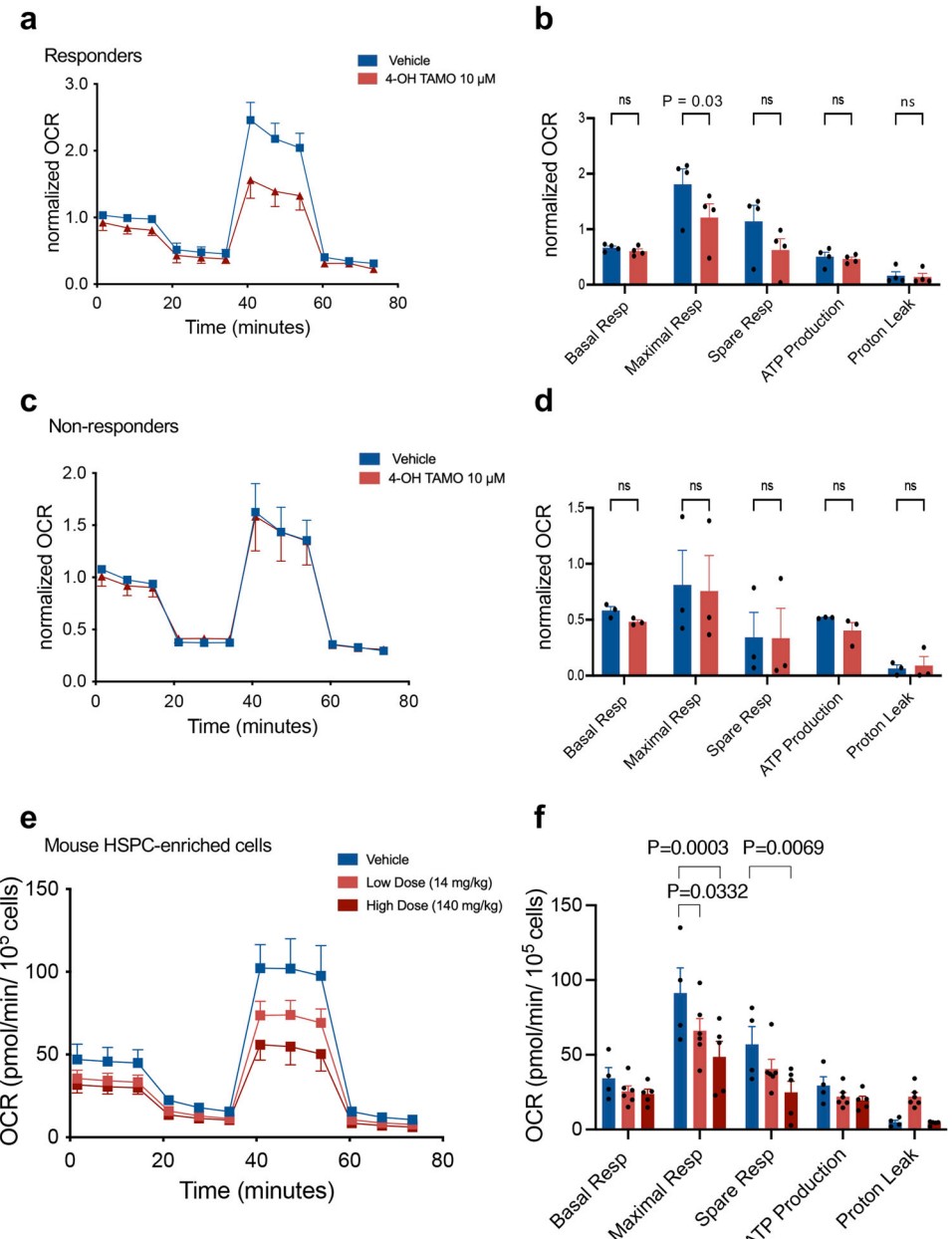

**Fig. 5 | Tamoxifen reduces mitochondrial respiration in blood cells from responders and HSPCs from mice treated with a dose comparable to study subjects.** **a**, **b** A high capacity of mitochondrial respiration in peripheral blood mononuclear cells (PBMCs) from responders is suppressed by 4OH-TAM (10 mM) treatment (*n* = 5 individuals). **c**, **d** Comparatively lower baseline OXPHOS and inhibition by 4OH-TAM (10 μM) in PBMCs from non-responders (*n* = 5 individuals). **a**–**d** Oxygen Consumption Rate (OCR) was measured in PBMCs after 24 h treatment with 4OH-TAM or vehicle and normalized to baseline. **e**, **f** Mitochondrial respiration in HSPC-enriched cells isolated from MPN mice treated with low dose tamoxifen (14 mg/kg, *n* = 6 animals), high dose tamoxifen (140 mg/kg, *n* = 5 animals) or vehicle (*n* = 4 animals) over two weeks (3 times/week). Note OXPHOS inhibition using a dose comparable to study patients. Data are means+SEM. *$p < 0.05$, **$p < 0.01$, ***$p < 0.001$, Two-way ANOVA and Dunnett's test.

# Methods
## Study design and oversight
The main goal of the TAMARIN study (EudraCT 2015-005497-38) was to obtain preliminary information on the safety and therapeutic activity of tamoxifen in MPN. TAMARIN is an investigator-driven A'herns design Phase II multicenter study approved by NHS Health Research Authority (IRAS 201126). The study was conducted in accordance with the Declaration of Helsinki and UK regulations (The Medicines for Human Use (Clinical Trials) Regulations 2004). The clinical trial authorization was provided by the Medicines and Healthcare products Regulatory Agency. The protocol was approved by the ethics committees of all involved institutions and is available with the full text of this article. The Clinical Study Protocol and the Statistical Analysis Plan, including the sample size calculations, interim safety analysis, analysis methods, Bayesian analysis, hematological response, and other exploratory outcomes are available in the Supp. Note. The authors designed the study, collected and analyzed the data, and wrote or edited the manuscript for submission.

## Human research participants
This study was approved by NHS Health Research Authority (IRAS 201126). The study was conducted in accordance with the Declaration of Helsinki and UK regulations (The Medicines for Human Use (Clinical Trials) Regulations 2004). The clinical trial authorization was provided

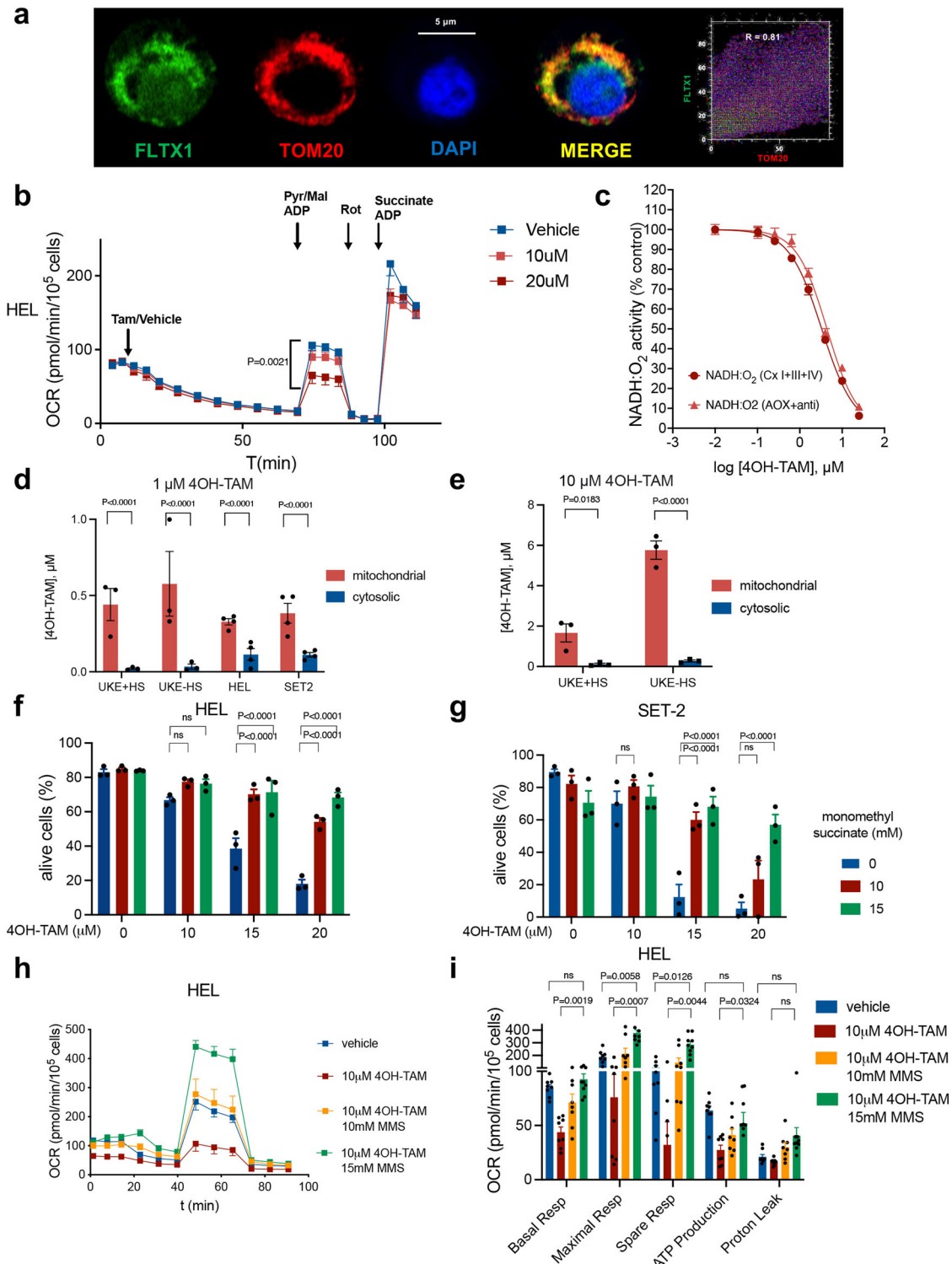

by the Medicines and Healthcare products Regulatory Agency. The study was registered on ISRCTN on 20-Jun-2016 (https://www.isrctn.com/ISRCTN65011803). Eligibility criteria included MPN patients aged ≥60 years, with eligible women being post-menopausal (defined as amenorrhoeic for at least 12 consecutive months following cessation of all exogenous hormonal treatments); confirmed diagnosis of *JAK2*-V617F, *CALR* 5 bp insertion (exon 9), or *CALR* 52 bp deletion (exon 9) positive ET, PV, or MF (primary or secondary) for ≥6 months. *JAK2*-V617F, *CALR* 5 bp insertion (exon 9), or *CALR* 52 bp deletion (exon 9) mutant allele burden ≥20% in peripheral blood granulocyte DNA at study entry (assessed via central review); WHO performance status

0–2. For patients with PV or ET, maintenance of platelet count ≤600 × 10⁹/L, WBC ≤ 25 × 10⁹/L and venesection requirements ≤1 per month for the previous 3 months prior to registration, without introduction of any new therapeutic agents for their MPN for 6 months prior to registration. For patients with MF, eligible patients showed no evidence of disease progression defined by IWG-MRT ELN criteria or new therapeutic agents introduced for a 6-month period before registration. Patients receiving cytoreductive therapy (with the exception of interferon alpha or investigational agents) for their MPN (not solely aspirin or venesection) were included, and this therapy was continued throughout. Adequate hepatic function, defined as bilirubin

**Fig. 6 | Tamoxifen reduces mitochondrial ATP generation by inhibiting respiratory complex I. a** Supra-resolution AiryScan2 confocal imaging of fluorescent tamoxifen derivative (FLTX1, green) colocalizing with the mitochondrial marker TOM20 (red) in HEL cells. Nucleus was stained with DAPI (blue). ImageJ plugin Colocalization Finder was used to generate a scatter plot of two selected channel intensity and calculate overlap coefficient for selected channels. R > 0.8 indicates significant colocalization. The analysis has been independently repeated three times, obtaining similar results. **b** 4OH-TAM dose-dependently inhibits mitochondrial respiration in permeabilized JAK2$^{V617F}$-mutant cells ($n = 10$ independent experiments). Average oxygen consumption rate (OCR) after treatment with plasma membrane permeabilizer, 4OH-TAM or vehicle, and 5 mM Pyruvate (Pyr), 2.5 mM malate and 1 mM adenosine diphosphate (ADP). **c** Dose-dependent inhibition of complex I NADH:O2 oxidoreduction rate by 4OH-TAM in absence or presence of antimycin (anti) to inhibit complex III ($n = 4$ independent experiments). **d**, **e** Dose-dependent accumulation of 4OH-TAM in the mitochondria of MPN cell lines 24 h after 1 mM (**d**) and 10 mM 4OH-TAM treatment (**e**). Note 4-fold higher mitochondrial 4OH-TAM concentration in sensitive (serum-deprived), compared with resistant (grown in competent medium) UKE-1 cells ($n = 3$ independent experiments). **f**, **g** Complex II activation with monomethyl succinate rescues JAK2$^{V617F}$-mutant HEL (**f**) and SET2 (**g**) cells from 4OH-TAM-induced cell death ($n = 3$ independent experiments). **h**, **i** OCR in HEL cells treated with 4OH-TAM or vehicle alone, or in combination with MMS ($n = 8$ independent experiments). The rescue of cell viability by complex II activation is explained by compensatory increase of mitochondrial respiration and ATP synthesis. **d-i** Data are mean ± SEM; *$p < 0.05$, **$p < 0.01$, ***$p < 0.001$, Two-way ANOVA and Dunnett's test.

≤1.5 × upper limit of normal (ULN) (patients with elevated bilirubin due to Gilbert's syndrome were eligible) or AST/ALT/ALP ≤ 2.5 x ULN, and adequate renal function (creatinine clearance >30 mL/min) were required. Male patients agreed to use effective contraception during participation in the trial and for 2 months after the last dose of trial treatment. All patients provided written informed consent. The Ethics statement and consent form included the use of human peripheral blood mononuclear cells for research.

Exclusion criteria included leukemic transformation (>20% blasts in blood, marrow or extramedullary site); accelerated phase of disease as indicated by ≥10% blasts in the peripheral blood); treatment of ET, PV, or MF with Interferon alpha or other investigational agents for their MPN within 6 months prior to trial entry (JAK inhibitors, such as ruxolitinib, were allowed if taken continuously for ≥6 months prior to registration); any of the following previous thrombotic events at any time: portal or other splanchnic venous thrombosis; vascular access complication; ischemia cerebrovascular; stroke; transient ischemic attack; superficial thrombophlebitis; venous thromboembolic events including pulmonary embolism (PE) and deep vein thrombosis (DVT); peripheral vascular ischemia; visceral arterial ischemia; acute coronary syndrome; myocardial infarction; previous malignancy within 5 years with the exception of adequately treated cervical carcinoma in situ or localized non-melanoma skin cancer; previous endometrial cancer, hyperplasia or polyps; prior treatment with hematopoietic stem cell transplantation; patients who did not carry JAK2-V617F, CALR 5 bp insertion (exon 9) or CALR 52 bp deletion (exon 9) mutations or whose allele burden was <20% at study entry (assessed via central review); female patients receiving hormone replacement therapy; hypertriglyceridemia > grade 1; any serious underlying medical condition (at the judgment of the Investigator), which could impair the ability of the patient to participate in the trial (e.g. liver disease, active autoimmune disease, uncontrolled diabetes, uncontrolled infection (HIV, Hepatitis B and C), known genetic defect (apart from MPN) relating to venous thromboembolic events, or psychiatric disorder precluding understanding of trial information); known hypersensitivity to tamoxifen or hypersensitivity to any other component of tamoxifen; concomitant drugs contraindicated for use with the trial drug according to the Summary of Product Characteristics; known planned scheduled elective surgery during study with the exception of dental and low risk eye surgery (e.g. cataracts).

The Trial Scheme for Eligibility and Central Analysis is summarized in Supp. Fig. 1.

## Protocol treatment
Tamoxifen was provided at the common dose used in ER+ breast cancer (20 mg oral daily, progressively escalated to 40 mg daily upon good tolerance and when there was no hematological response or mutant allele burden reduction at 12w). All patients received trial treatment for 24 weeks. Treatment continuation was encouraged but not mandated after 24w for patients who did not experience persistent side effects greater than grade 1 or thrombotic events of any grade and that fulfilled one or more of the following criteria at 24 weeks: ≥25% reduction in allele burden compared to baseline; improvement of hematological response compared to baseline without changes in cytoreductive therapy dose according to 2009 ELN criteria for ET/PV patients and to IWG-MRT response criteria for MF patients; a decrease in requirement for cytoreduction without deterioration of hematological response compared to baseline according to 2009 ELN criteria for ET/PV patients and to IWG-MRT response criteria for MF patients. Their response was reassessed after 36 and 48 weeks of treatment as applicable. The Study Protocol is available in the Supp. Note.

## Design, quantification, and statistical analysis
The original sample size target of 42 was based on an A'Herns design and power of 80%. An alternative approach adopted a Bayesian framework to adjust to the total 38 patients recruited. A beta-binomial conjugate analysis confirmed an equal probability with the original A'Herns design and the Bayesian framework considering the 38 patients recruited and concluded that the primary outcome would need to be observed in at least 3 patients to meet the success criteria. The primary endpoint was a reduction in the peripheral blood JAK2V617F, CALR 5 bp insertion (exon 9) or CALR 52 bp deletion (exon 9) mutant allele burden of ≥50% at 24 weeks. Under the A'hern design, it was necessary to observe at least 3 successes (i.e. reductions in allele burden of ≥50% in the 38 patients recruited). Secondary outcomes were the proportion of patients with a reduction in the peripheral blood JAK2-V617F, CALR 5 bp insertion (exon 9), or CALR 52 bp deletion (exon 9) mutant allele burden of ≥50% at 12 weeks; toxicity measured as the number of grade 3 and 4 adverse events reported; the number of thrombotic events of any grade reported and validated; duration of hematological response calculated as time from registration to progression for patients who entered the study in response (CR or PR). For patients who entered the trial in stable disease, the time between first recorded response to the date of progression. Progression was defined as loss of response for PV/ET patients and evidence of disease progression for MF patients. PV/ET patients who continued to achieve a response, or MF patients who had no evidence of disease progression at the end of the trial were censored at date last seen. Hematological response was defined according to 2009 ELN criteria for ET/PV patients[62] and no evidence of disease progression for MF patients according to IWG-MRT response criteria;[63] proportion of patients in each response category according to IWG-MRT response criteria[63] for MF patients and 2013 ELN response criteria[64] for ET/PV patients at 24 weeks of treatment; proportion of patients showing an improvement in response category at 24 weeks compared to baseline according to 2009 ELN criteria for ET/PV patients[62] and according to IWG-MRT response criteria[63] for MF patients; patients who are in a higher category at week 24 compared to baseline were classed a success; patients who enter the trial in CR and who maintain a CR were classed as a success in this outcome. Exploratory outcomes included

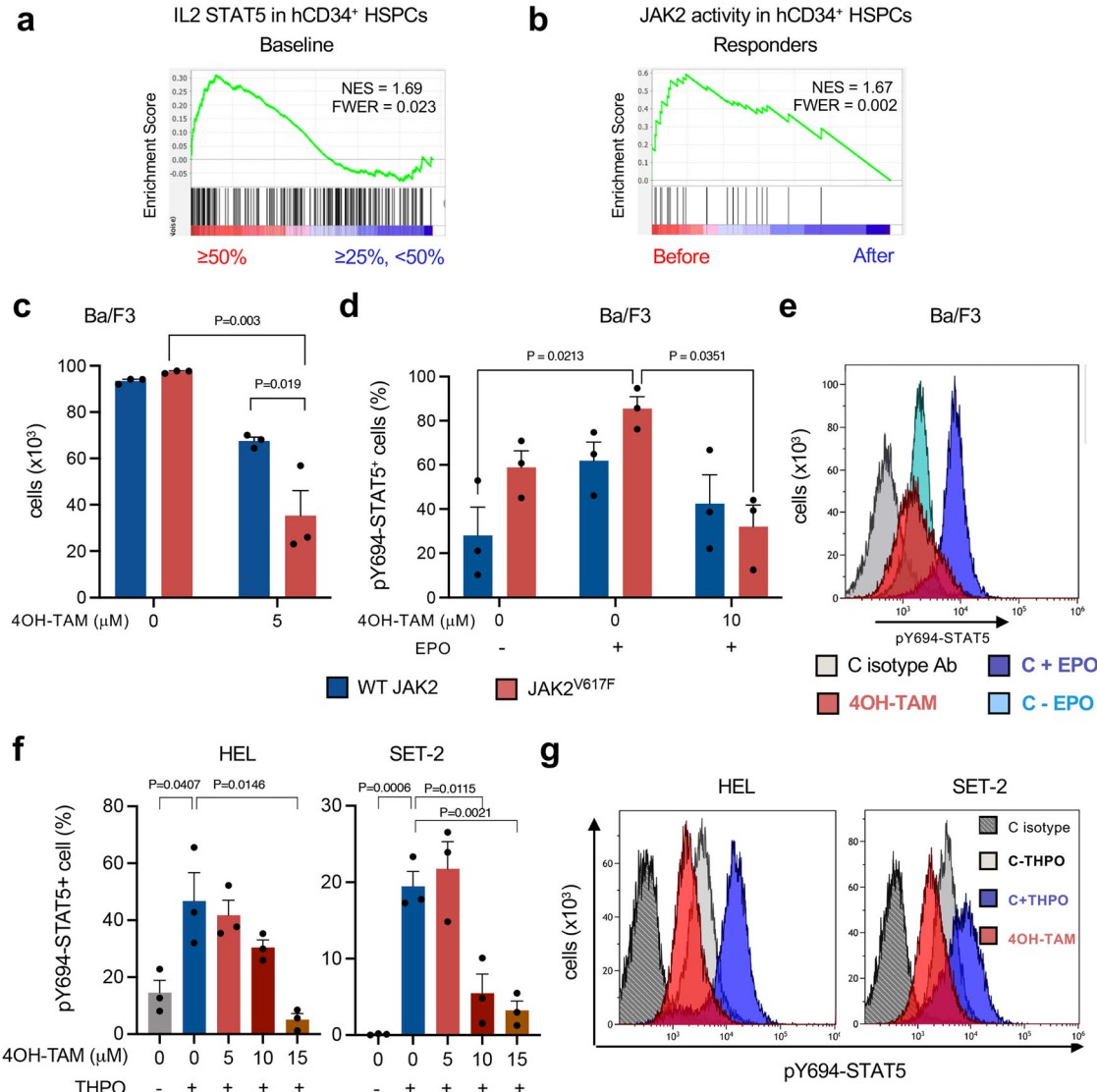

**Fig. 7 | Tamoxifen preferentially inhibits oncogenic JAK-STAT signaling. a** Gene set enrichment analysis (GSEA) of HSPCs obtained at baseline from study patients showing allele burden reductions ≥50% (*n* = 3 individuals) or between ≥25% and 50% (*n* = 3 individuals) 24w after tamoxifen treatment. Interleukin-2 (IL-2)-STAT5 signatures were enriched in HSPCs from patients achieving ≥50% allele burden reductions. **b** Selective downregulation of JAK2-related genes in HSPCs from study patients showing allele burden reductions ≥50% (*n* = 3 individuals) or between ≥25% and 50% (*n* = 3 individuals) 24w after tamoxifen treatment. NES, normalized enrichment score. FWER, family-wise error rate. **c–e** Mouse Ba/F3 cells expressing the erythropoietin (Epo) receptor[44] and transduced with mutant JAK2[V617F] or WT JAK2[45] were treated with 4OH-TAM or vehicle. **c** Ba/F3 cells carrying *JAK2[V617F]* are twice as sensitive to 4OH-TAM-induced cell death compared with cells expressing WT JAK2 (*n* = 3 independent experiments). Data are mean + SEM.

**d, e** Selective inhibition of oncogenic JAK-STAT signaling by 4OH-TAM in Ba/F3 cells. **d** Frequency of WT/mutant JAK2-expressing Ba/F3 cells with phosphorylated (p) STAT5 at residue Y694 at baseline or after EPO (5 U/ml) stimulation, in combination with 4OH-TAM (red) or vehicle (blue). Data are mean + SEM.
**e** Representative flow cytometry histograms of pSTAT5 level at baseline (light blue) or after EPO (5 U/ml) stimulation (dark blue), in combination with 4OH-TAM (red) or vehicle (*n* = 3 independent experiments). **f, g** 4OH-TAM dose-dependently inhibits oncogenic JAK-STAT signaling in *JAK2[V617F]*-mutated human cells. Frequency of pSTAT5[+] HEL cells or SET-2 cells (**f**) and representative flow cytometry histograms of pSTAT5 level (**g**) at baseline or after thrombopoietin (THPO, 100 ng/ml) stimulation (blue), in combination with 4OH-TAM (red) or vehicle (*n* = 3 independent experiments). C, control. **c-d, f** Data are mean ± SEM; *p* < 0.05, **p* < 0.01, ****p* < 0.001, One-way ANOVA and Tukey's test.

the proportion of patients showing a decrease in allele burden at 12 and 24 weeks compared to baseline was presented as the number of patients who have shown a decrease of any amount between baseline and week 12 or between baseline and week 24; proportion of patients showing a decrease in requirement for cytoreduction therapy at 24 weeks compared to baseline; proportion of patients showing a decrease in allele burden of ≥50% at 36 and 48 weeks compared to baseline; duration of reduction in the peripheral blood *JAK2*-V617F, *CALR* 5 bp insertion (exon 9), or *CALR* 52 bp deletion (exon 9) mutant allele burden, defined as time from first observed reduction of ≥50% until reduction from baseline becomes <25%; RNAseq studies on

CD34+ HSPCs isolated from peripheral blood at different time points throughout the study.

### Statistics and reproducibility

For the exploratory outcomes, statistical analysis was performed with GraphPad Prism 9. *P* < 0.05 was considered statistically significant. All data of laboratory experiments are presented as mean ± SEM. Prism software (GraphPad Software) was used for all statistical analyses. Measurement was taken from different samples with at least three biological replicates. No statistical method was used to predetermine sample sizes. No data were excluded from the analyses. The

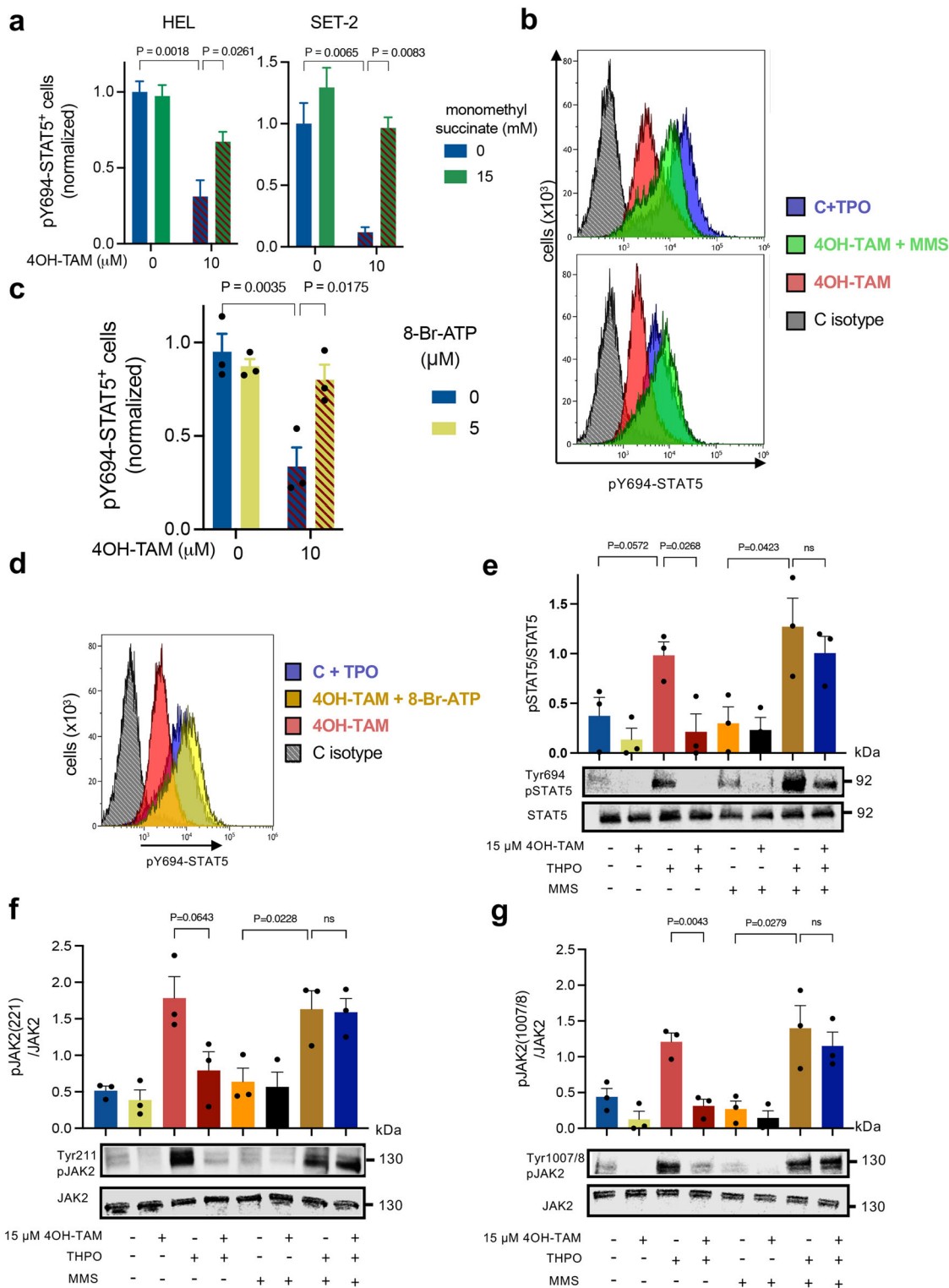

experiments were not randomized. The investigators were not blinded to allocation during in vitro or animal experiments and outcome assessment.

## Cell culture

The MPN cell lines HEL, UKE-1, and SET2 were obtained from the American Type Culture Collection (ATCC; http://www.atcc.org, Manassas, VA). The MARIMO cell line was kindly provided by Dr Juan Li (Wellcome-MRC Cambridge Stem Cell Institute). The estrogen receptor-positive breast cancer MCF-7 were provided by Dr Jason Carroll (Cancer Research UK). HEL, SET2 and MARIMO cells were cultured in phenol red-free RPMI-1640 medium (Thermo Fisher Scientific, Cat. No. 32404014) supplemented with 10% Charcoal Stripped fetal bovine serum (Thermo Fisher Scientific, Cat. No. 12676029) and antibiotics (penicillin, 100 I.U. /ml; streptomycin, 100 μg/ml). UKE-1 cells were cultured in phenol red-free IMDM medium (Thermo Fisher Scientific, Cat. No. 21056023) supplemented with 10% Charcoal Stripped fetal bovine serum, antibiotics (penicillin, 100 I.U./ml; streptomycin, 100 μg/ml), 10% heat-inactivated horse serum (Thermo Fisher Scientific, Cat. No. 26050070) and 1 μM hydrocortisone (Sigma-Aldrich, Cat.

**Fig. 8 | Inhibition of pathogenic JAK-STAT signaling by tamoxifen is rescued by complex II activation or ATP supplementation. a-b** Frequency of pSTAT5⁺ HEL cells or SET-2 cells (**a**) and representative flow cytometry histograms of pSTAT5 expression (**b**) at baseline or after thrombopoietin (THPO) stimulation (blue), in combination with 4OH-TAM (red) or vehicle (*n* = 3 independent experiments). Combined treatment with the mitochondrial complex II substrate monomethyl succinate (MMS, green) rescues the reduction of pSTAT5 by 4OH-TAM. C, control. **c-d** Frequency of pSTAT5⁺ THPO-stimulated HEL cells (**c**) and representative flow cytometry histograms of pSTAT5 expression (**d**) in HEL cells treated with 4OH-TAM alone (red) or vehicle (blue), or in combination with cell-permeable ATP (5 mM 8-Br-ATP, yellow) (*n* = 3 independent experiments). ATP supplementation rescues the reduction of pSTAT5 by 4OH-TAM. **a,c** Data are mean + SEM; *$p < 0.05$, **$p < 0.01$, ***$p < 0.001$, One-way ANOVA and Tukey's test. **e** Quantification (top) and representative Western blots (bottom) of STAT5 and phosphorylated (p) STAT5 level at

baseline or after thrombopoietin (THPO) stimulation, in combination with 4 h vehicle/4OH-TAM treatment and mitochondrial complex II substrate monomethyl succinate (10 mM MMS) in HEL cell lines. Combined treatment with the mitochondrial complex II substrate monomethyl succinate (MMS) rescues the reduction of THPO-induced pSTAT5 by 4OH-TAM (*n* = 3 independent experiments). **f-g** Quantification (top) and representative Western blots (bottom) of JAK2 and phosphorylated (p) JAK2 (**f**, Tyr221; **g**, Tyr1007/8) level at baseline or after thrombopoietin (THPO) stimulation, in combination with 4 h vehicle/4OH-TAM treatment and monomethyl succinate (10 mM MMS) in HEL cell lines, showing the reduction of thrombopoietin-induced JAK2 phosphorylation by 4OH-TAM could be rescued by the addition of succinate in human *JAK2^V617F*-mutated cell lines (*n* = 3 independent experiments). **e-g** Data are mean + SEM; *$p < 0.05$, **$p < 0.01$, ***$p < 0.001$, One-way ANOVA and Dunnett's test. **e-g** Data are mean + SEM; *$p < 0.05$, **$p < 0.01$, ***$p < 0.001$, One-way ANOVA and Dunnett's test.

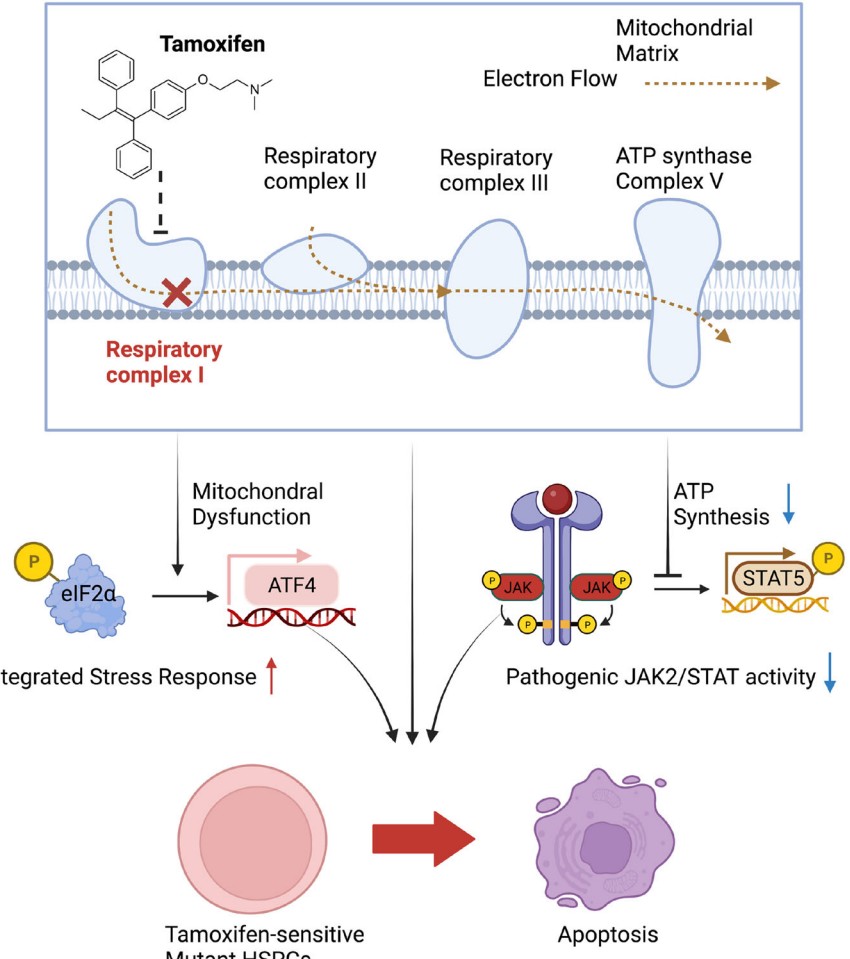

**Fig. 9 | HSPCs with high JAK-STAT signaling, integrated stress response and mitochondrial respiration are particularly sensitive to tamoxifen-induced cell death.** Tamoxifen derivatives accumulate in the mitochondria, where they bind to and inhibit complex I. Biochemical complex I inhibition adds to reduced complex I gene transcription after tamoxifen treatment. Additionally, tamoxifen further increases integrated stress response in HSPCs and tips the balance from protective

(p-eIF2α-dependent) to proapoptotic (ATF4-mediated) unfolded protein response. Through transcriptional regulation and previously unrecognized direct metabolic effects, tamoxifen can eliminate cells with pathogenic JAK-STAT activation by modulating integrated sress response and inhibiting mitochondrial complex-I and ATP generation, preventing pathogenic STAT5 phosphorylation.

No. H0888). MCF-7 cells were cultured in DMEM medium (Thermo Fisher Scientific, Cat. No. 10569010) supplemented with 10% heat inactivated fetal bovine serum (Thermo Fisher Scientific, Cat. No. 26140079) and antibiotics (penicillin, 100 I.U. /ml; streptomycin, 100 μg/ml). Ba/F3 cells were cultured in phenol red-free RPMI medium supplemented with 10% Charcoal Stripped fetal bovine serum, antibiotics (penicillin, 100 I.U./ml; streptomycin, 100 μg/ml), and 10 ng/ml recombinant murine IL-3 (PeproTech, Cat. No. 213-13).

Cells were cultured at 37 °C with 5% CO₂ in a water-jacketed incubator.

## Mouse model

Age-matched, wild-type female C57BL/6 J mice (Charles River Laboratories) were used for in vivo treatments and as recipients of bone marrow transplantation assays to generate MPN model, as previously described[22]. Mice were housed in specific pathogen-free facilities. All

experiments using mice followed protocols approved by the Animal Welfare Ethical Committees (AWERB) at the University of Cambridge (PPL P0242B783). All experiments were compliant with UK regulations.

MPN was induced by transplantation of $2*10^6$ bone marrow nucleated cells from *Vav-iCre;JAK2*$^{V617F}$ mice[65] into lethally irradiated C57BL/6 J recipients. MPN phenotype was confirmed by high circulating platelet (over $1.5*10^9$ platelets/ml blood) in recipient mouse. The humane endpoints considered were any of these: animal subdued (not moving and only responsive to gentle stimulus); animal losing 15% of basal body weight; animal losing 10% of basal body weight and shows piloerection, inactivity, or a hunched posture as a sign of pain. For short treatments (2 weeks), mice were injected i.p. with 14 or 140 mg/ kg body weight of tamoxifen (Sigma-Aldrich, Cat. No. T5648) or vehicle (corn oil, Sigma-Aldrich, Cat. No. C8267), three times per week on alternate days. Mouse lineage-negative hematopoietic cell population was collected by immunomagnetic depletion of lineage-committed cells from mouse bone marrow (Mouse Hematopoietic Progenitor (Stem) Cell Enrichment Set, BD Bioscience, Cat. No. 558451).

## RNAseq

Laboratory correlative studies investigated possible biomarkers of response based on the transcriptome of peripheral blood CD34+ HSPCs. After ficoll centrifugation of blood, mononuclear cells were subjected to red blood cell lysis and CD34+ HSPCs were immuno-magnetically isolated using the Ultrapure CD34 microbead kit (Miltenyi Biotec, Cat. No. 130-100-453) suing AUTOMACS (Miltenyi Biotec). RNA was extracted using Trizol (Invitrogen, Cat. No. 15596026). Briefly, the Smart-seq2 protocol was implemented with improved reverse transcription, template switching and preamplification to increase both yield and length of cDNA libraries generated from individual cells. The Illumina-compatible cDNA sequencing library was prepared with Nextera XT DNA Library Preparation Kit (Illumina, Cat. No. FC-131-1096). The quality, quantity, and the size distribution of the Illumina libraries were determined using the DNA-1000 Kit (Agilent Bioanalyzer, Cat. No. 5067-1504). Libraries were sequenced on the Novaseq 6000 (Illumina). Fastq files containing reads for each library were extracted and demultiplexed using Casava v1.8.2 pipeline. Sequencing adapter contaminations were removed from reads using cutadapt software tool (MIT) and the resulting reads were mapped and quantified on the transcriptome using RSEM v1.17[66]. Only genes with more than 2 counts per million in at least 2 samples were considered for statistical analysis. The RNAseq data from CD34+ HSPCs of study subjects has been deposited in GEO under the Accession Number GSE172022.

For RNA-Seq of *JAK2*$^{V617F}$-mutated human cells lines, HEL cells and UKE-1 cells cultured with or without horse serum (UKE-1 +/-HS respectively) were treated with 10 μM tamoxifen or vehicle for 4 h and RNA was extracted from tamoxifen-treated cells by Trizol (Invitrogen, Cat. No. 15596026) and cleaned-up with RNeasy Mini Columns (Qiagen, Cat. No. 74104). RNA-Seq libraries were prepared from total RNA using poly(A) enrichment of the mRNA and sequenced by HiSeq 4000 (Illumina). The RNAseq data from HEL cells and UKE-1 cells has been deposited in GEO under the Accession Number GSE172023.

Differential expression analysis was performed using DESeq2 package[67]. Gene-set enrichment analyses (GSEA) was performed against the collection of curated gene sets available in the Molecular Signatures Database (http://www.broadinstitute.org/gsea/msigdb/index.jsp), including KEGG, Biocarta and Reactome pathways as well as a collection of gene expression signatures associated with chemical or genetic perturbations. Significance of gene set enrichment between the two conditions was assessed with GSEA software as previously described[68] (http://www.broadinstitute.org/gsea/index.jsp), using a weighted statistic, ranking by signal to noise ratio and 1000 gene-set permutations. GSEA output using GO dataset was prepared for Integrative pathway as previously described[69]. The node cut-off for q-value is 0.005 and 0.1 for responder and non-responder network

respectively. Integrated pathway enrichment map is visualized by Cytoscape[70].

## Colony-forming unit assay

Human peripheral blood mononuclear cells (PMNCs) were cultured in StemSpan™ Hematopoietic Cell Media (Stemcell Technologies, Cat. No. 100-0073) supplemented with StemSpan™ CC100 cytokine cocktail (Stemcell Technologies, Cat. No. 2690), 5mM L-glutamine and antibiotics (penicillin, 100 I.U. /ml; streptomycin, 100 μg/ml). The cells were seeded at a density of $1 × 10^6$ cells/ml and treated with 4OH-TAM (10 μM) (Sigma-Aldrich, Cat. No. H6278) or vehicle for 24 h. To perform CFU assay, PMNCs were plated in StemMACS HSC-CFU complete with Epo (Miltenyi Biotec, Cat. No. 130-091-280) at a density of $5 × 10^4$ cells/ ml. Cultures were incubated at 37 °C for 14 days.

## Flow cytometry

For measurement of pSTAT5, cell lines were washed and replated at $10^6$ cells/ml in basal media (phenol red-free IMDM or RPMI) without serum for 2 h. Cells were treated with 4OH-TAM (5 μM, 10 μM and 15 μM) or vehicle for 45 min and were subsequently stimulated with recombinant human THPO (100 ng/ml) (Peprotech, Cat. No. 300-18) or EPO (5 U/ml) (Peprotech, Cat. No. P8783) for 15 min. Cells were washed with PBS and fixed in 2% paraformaldehyde at 37 °C for 10 min, centrifuged, washed once in p-STAT staining buffer (PBS, pH 7.2, with 0.2% BSA and 0.09% sodium azide) and permeabilized in 100% ice cold methanol on ice for 30 min as previously described[71]. Cells were washed twice with pSTAT staining buffer and labeled with anti−pY694-STAT5 (BD Biosciences, clone 47, Cat. No. 562076, RRID: AB_11154412) or isotype control (BD Biosciences, clone MOPC-21, Cat. No. 557732, RRID: AB_396840) for 20 min. Cells were washed once in pSTAT staining buffer.

To measure apoptosis, cell lines were seeded at $10^6$ cells/ml in growth medium and treated with 4OH-TAM (5 μM, 10 μM, 15 μM, and 20 μM) or vehicle. After 24 h and 48 h incubation, $2×10^5$ cells were harvested, washed with Annexin V binding buffer (140 mM NaCl, 4 mM KCl, 0.75 mM MgCl2, and 10 mM HEPES in bidistilled water, pH 7.4) and labeled using the FITC Annexin V antibody (BioLegend, Cat. No. 640906, RRID: AB_2561292, 1:100) and DAPI (Thermo Fisher Scientific, Cat. No. D1306). The fraction of live cells was determined by selecting Annexin$^-$ DAPI$^-$ cells. Cells were analyzed with a Gallios Flow Cytometer using Kaluza software (BD Biosciences).

## Quantitative RT-PCR

To detect the mutation type in patient HSPCs, DNAs extracted using the QiampDNA mini kit (Qiagen, Cat. No. 51304) from patient granulocyte samples were standardized to 20 ng/ul and all samples tested at each time-point. Samples were excluded if they didn't meet acceptance criteria of more than 60% mature granulocytes in DNA preparation, cell viability at freezing <95%, and less than 36 h in transit. A repeat blood sample was requested in these cases. JAK2V617F mutation analysis was performed as previously described[72]. A similar mutation-specific primer approach was used for CALR 52 bp deletions, and quantified against the relative amount of product of CALR exon 9/ intron 9 with primers lying outside the mutated region of the gene. All samples were run in triplicate and products were normalized to the relevant plasmid standard and the test mean expressed as a percentage of the control region mean value. Primer sequence for CALR wild-type and 5 bp insertion is 5′ GCAGCAGAGAAACAAATG 3′ (forward) and 5′ GCCTCTCTACAGCTCGTCCTT 3′ (reverse). Reverse primer for CALR 52 bp deletion is same with CALR wild-type and 5 bp insertion but the specific forward primer for CALR 52 bp deletion is 5′ ACAGGACGAG-GAGCAGAGAAC 3′. The probe sequence for wild-type and CALR 52 bp deletion is 5′−6FAM-TGAGGATGAGGAGGATGAGG-BHQ1 −3′. The sequence of specific probe for CALR 5 bp insertion is 5′ 6FAM − GTC[ + C]TCATCATCCT[ + C]CTT - BHQ1 3′.

The expression levels of nuclear estrogen receptors and membranal receptor in MPN cell lines were determined by qRT-PCR using standard methodologies. Total RNA was extracted using the RNeasy Mini Kit (Qiagen, Cat. No. 74104), and 500 ng was used in a reverse transcriptase reaction (High-Capacity cDNA Reverse Transcription kit, Applied Biosystems, Cat. No. 4368814) to generate cDNA for the template in quantitative real-time PCR reactions, which was performed using the PowerUp SYBR Green Master Mix (Applied Biosystems, Cat. No. A25742). Expression level was normalized to expression of endogenous control gene GAPDH. Primer sequence for human ESR1 is 5′ TGGGCTTACTGACCAACCTG 3′ (forward) and 5′ CCTGATCATG-GAGGGTCAAA 3′ (reverse); for human GAPDH is 5′ GTCTCCTCTG ACTTCAACAGCG 3′ (forward) and 5′ ACCACCCTGTTGCTGTAGCCAA 3′ (reverse).

### ATF4, XBP1s reporter cell line generation

HEL-ATF4 and UKE-1-ATF4 were obtained by lentiviral infection of HEL and UKE-1 cells respectively. pLVX-ATF4 mScarlet NLS (Addgene plasmid # 115969) were gifts from David Andrews[73]. HEK293T cells were transfected with the pLVXATF4 mScarlet vectors and the viral particles were collected in culture medium supernatant as described[74]. To generate stable HEL and UKE-1 cells expressing reporters, HEL and UKE-1 were infected by supernatant containing viral particles and selected with 2 mg/mL of Puromycin (Thermo Fisher Scientific, Cat. No. A1113803) for one week. For ATF4 12-h monitoring experiments, $10^6$ stably transfected cells were seeded in a 6-well plate with vehicle, thapsigargin (Tocris Bioscience, Cat. No. 1138) and 4OH-TAM. Cells were collected at specific time point (2H, 4H, 8H and 12H) and labeled with DAPI. The reporter fluorescence level was determined by selecting DAPI- cells.

### Immunoblotting

For the immunoblotting of eIF2α phosphorylation and selective ATF4 translation, cells were treated with 4OH-TAM (10 μM, 15 μM and 20 μM) for 8 h or with 10 μM for 2, 4 and 8 h. Whole cell lysate was extracted from the RIPA lysis solution (Abcam, Cat. No. ab156034) with the Complete Protease Inhibitor Cocktail Tablets (Roche Applied Science, Cat. No. 11697498001) and PhosSTOP phosphatase inhibitor (Roche Applied Science, Cat. No. 4906845001). Equal amounts of protein (50 μg) were boiled for 5 min in 1 × Laemmli sample buffer (BioRad, Cat. No. 1610747) and separated on Mini-PROTEAN precast gels (BioRad, Cat. No. 4569034). The binding of antibody (ERα Antibody (Santa Cruz, Cat. No. sc-8002, clone F-10, RRID: AB_627558), Phospho-eIF2α XP antibody (Ser51) (Cell Signaling Technology, Cat. No. 3398, clone D9G8, RRID: AB_2096481), eIF2α antibody (Cell Signaling Technology, Cat. No. 5324, clone D7D3, RRID: AB_10692650), ATF-4 antibody (Cell Signaling Technology, Cat. No. 11815, clone D4B8, RRID: AB_2616025), Phospho-Stat5 Antibody (Thermo Fisher Scientific, Cat. No. 71-6900, RRID: AB_2533991), Stat5 antibody (BD Biosciences, Cat. No. 610191, RRID: AB_397590), JAK2 antibody (Thermo Fisher Scientific, Cat. No. AHO1352, clone 691R5, RRID: AB_2536334), Phospho-JAK2 (Tyr1007/1008) Antibody (Cell Signaling Technology, Cat. No. 3771, RRID: AB_330403), Phospho-JAK2 (Tyr221) Antibody (Cell Signaling Technology, Cat. No. 3774, RRID: AB_390750), Anti-VDAC1/Porin Antibody (Santa Cruz, Cat. No. sc-390996, clone B-6, RRID: AB_2750920) and α-Tubulin antibody (Sigma-Aldrich, Cat. No. T5168, clone B-5-1-2, RRID:AB_477579)) was detected using an enhanced chemiluminescence horseradish peroxidase (HRP) substrate (Thermo Fisher Scientific, Cat. No. 32106). All primary antibodies were used at a dilution of 1:1000, expect α-Tubulin antibody, which was used at a dilution of 1:5000. Imaging and band quantification were carried out using Odyssey Fc Imaging System and Image J/Fiji Software.

### Metabolic studies

Seahorse XFp Cell Mito Stress Tests were performed using Seahorse XF Cell Mito Stress Test Kit (Agilent Technologies, Cat. No. 103015-100) according to the manufacturer's protocol on Seahorse XFe96 Analyzer (for cell lines) and Seahorse XF HS Mini Analyzer (for mouse bone marrow lineage-negative cells and human peripheral blood mononuclear cells) (Agilent Technologies). To perform the metabolic assays on cell lines, Seahorse XFe96/XF Pro Cell Culture Microplates (Agilent Technologies, Cat. No. 103794-100) were coated with Cell-Tak (Corning, Cat. No. 354240) in 0.1 M sodium bicarbonate solution (pH 7.4) for 20 min at room temperature, washed twice with water and stored at 4 °C one day before the assay. Cells were treated with 4OH-TAM for 2 h and washed with XF Mito Stress medium (Agilent XF DMEM Medium (Agilent Technologies, Cat. No. 103335) with 10 mM glucose, 1 mM pyruvate, and 2 mM glutamine, pH 7.4). Cells were then maintained in XF Mito Stress medium at 37 ∘C in an incubator for 1 h to pre-equilibrate. For mouse bone marrow lineage-negative cells and human peripheral blood mononuclear cells, cells are seeded in Seahorse XFp PDL Cell Culture Miniplates (Agilent Technologies, Cat. No. 103722-100). In Cell Mito Stress Tests, the concentrations for oligomycin, FCCP and rotenone/antimycin A are 1 μM, 0.5 μM, and 0.5 μM respectively.

For the XF Real-Time ATP Rate Assay (Agilent Technologies, Cat. No. 103592-100), cells were washed with the ATP assay medium (Agilent XF DMEM Medium with 10 mM glucose, 1 mM pyruvate, and 2 mM glutamine, pH 7.4). In Real-Time ATP Rate Assay, the concentrations for oligomycin and rotenone/antimycin A are 1.5 μM and 0.5 μM respectively. Data analysis was performed using the Seahorse XFe Wave software and statistics were analyzed in Prism 7.0.

For permeabilization assay, cells were washed and maintained in pre-warmed mannitol and sucrose buffer (MAS, pH 7.2) (220 mM mannitol, 70 mM sucrose, 10 mM KH2PO4, 5 mM MgCl2, 2 mM HEPES, 1 mM EGTA, 4 mg/ml BSA), following by adding 1 nM XF Plasma Membrane Permeabilizer (PMP) (Agilent Technologies, Cat. No. 102504-100) before running the assay. After the first injection of tamoxifen and vehicle control, mitochondrial activity was initiated by pyruvate (5 mM), malate (2.5 mM), and ADP (1 mM) for measuring complex I-driven respiration, followed by succinate (10 mM) and ADP (1 mM) for complex II-driven respiration.

### Confocal imaging

Cells were fixed in fixation solution (PBS solution with 2% paraformaldehyde, 0.1% glutaraldehyde, 150 mM sucrose) for 30 min at room temperature and then permeabilized in permeabilization solution (PBS with 0.1% Triton-X-100). After incubation of TOM20 antibody (Santa Cruz, Cat. No. sc-17764, clone F-10, RRID: AB_628381, 1:100), samples were treated with Image-iT™ FX signal enhancer (Thermo Fisher Scientific, Cat. No. I36933) for 30 min, followed by two-hour incubation of 50 μM FLTX1 (MedChemExpress, Cat. No. HY-119437) at room temperature[35,36]. Cells were mounted by ProLong Diamond Antifade Mountant (Thermo Fisher Scientific, Cat. No. P36961) and visualized by LSM 980 Airyscan 2 (Zeiss). Colocalization analysis was performed by ImageJ plugin Colocalization Finder (https://imagej.nih.gov/ij/plugins/colocalization-finder.html).

### Respiratory complex I kinetic assays

Bovine heart mitochondrial membranes were prepared as described previously[75]. Assays were performed at 32 °C in 10 mM Tris-SO₄ 250 mM sucrose (pH 7.2) using 200 μM NADH in the presence of 1.5 μM horse heart cytochrome $c$ (Sigma-Aldrich, Cat. No. C7752) and quantified by the absorbance of NADH ($\varepsilon 340{-}380 = 4.81$ mM$^{-1}$·cm$^{-1}$)[75] in 96-well plates using a Molecular Devices Spectramax 384 plus platereader with Softmax Pro software. Catalysis was initiated by the addition of NADH, and rates reported are for the linear regression of the maximal rates. To bypass complexes III and IV, assays were performed in the presence of 1 μM Antimycin (Sigma-Aldrich, Cat. No. A8674) and 15 μg/mL alternative oxidase (AOX), which was prepared as described previously[76]. Hydroxytamoxifen was added to the assays from DMSO stocks, and compared to appropriate DMSO controls.

## Mitochondrion Isolation

Mitochondria isolation was performed using the commercial kit (Mitochondria Isolation Kit for Cultured Cells, Thermo Fisher Scientific, Cat. No. 89874) according to the manufacturer's protocol. In brief, 50 million cells pre-treated with 4-OH tamoxifen for 24 h were harvested and collected in reagent A, followed by 2-min incubation on ice. Cells were then treated with 10 µL of reagent B and incubated for 5 min on ice with vertexing every minute. After adding the reagent C, homogenized cells were centrifuged at 700 g for 10 min at 4 °C and the pellet was discarded. The crude fraction was centrifuged at 12,000 ×g for 15 min at 4 °C. While the supernatant was stored as cytosolic fraction, the pellet containing the mitochondria-enriched fraction was washed with reagent C and centrifuged at 12,000×g for 5 min at 4 °C. T

## 4OH-tamoxifen measurement

4-hydroxytamoxifen was separated using revrse phase LC separation with Acquity UPLC column (50 mm×2.1 mm) T3 (C18) bonding and endcapping, packed with high strength silica (HSS) particle substrate. Sample volume of 5 µL was injected using Simadzu Nexera ultra-high performance liquid chromatography (UHPLC) (Shimadzu, Japan) coupled with Q-trap SCIEX 6500 (QQQ) mass spectrometer (SCIEX). The column oven temperature was maintained at 40 °C, with a flow rate of 0.4 mL/min. A linear gradient was applied ranging for 5.0 min ramping from 30% to 95% solvent (0.1% formic acid in acetonitrile). MS parameters for data acquisition using SCIEX 6500 triple quadropole system included: positive Electrospray ionization (ESI); Ion source voltage at 5500 V; Source temperature at 550 °C; collision activate dissociation (CAD) at 8; Nebulizer gas (GS1) at 50; and Auxillary gas (GS2) at 60. 4-hydroxytamoxifen parent ion was monitored (major precursor [M-H]+ ions at m/z 388.0) and daughter ion (major fragmented product ion at m/z 72.2).

## Reporting summary

Further information on research design is available in the Nature Portfolio Reporting Summary linked to this article.

## Data availability

The study protocol is provided as Supplementary Note. The Statistical Analysis Plan, including the sample size calculations, interim safety analysis, analysis methods, and Bayesian analysis are available in the "Methods" section. The primary outcomes, secondary outcomes, and exploratory outcomes are available in the figures, supplementary figures, and supplementary tables. The CRCTU is committed to responsible and controlled sharing of anonymized clinical trial data with the wider research community to maximize potential patient benefit while protecting the privacy and confidentiality of trial participants. Data anonymized in compliance with the Information Commissioners Office requirements, using a procedure based on guidelines from the Medical Research Council (MRC) Methodology Hubs, will be available for sharing with researchers outside of the trials team within 12 months of the primary publication. Data Sharing requests can be submitted to the CRCTU and will be reviewed by the CRCTU Directors Committee and relevant Trial Management Group (www.birmingham.ac.uk/research/crctu). The RNAseq data from CD34+ HSPCs of Study subjects has been deposited in GEO under the Accession Number GSE172022. The RNAseq data from HEL cells and UKE-1 cells has been deposited in GEO under the Accession Number GSE172023. Additional data supporting the findings in this work are available in the main text, figures, supplementary figures, supplementary tables, and supplementary note. Source data are provided with this paper.

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

## Acknowledgements

We thank the patients and their families for making this study possible; the support by Blood Cancer UK under the Trials Acceleration Programme (TAP) and the Trial Steering Committee members T. Sommervaille, S. Ali and J. Lambert for study design, data collection, and analysis; C. Fielding, H. Matsubara, C. Bernardo-Castiñeira, A. Sommerschield and members of the S.M.-F. group for help and discussions; J. Carroll, B. Huntly, J. Li, and A.R. Green for critical advice and support; Cambridge NIHR BRC Cell Phenotyping Hub for technical assistance; D. Bexheti and R. Houghton (Cancer Research UK) for LC-MS/MS. Samples were provided by the Cambridge Blood and Stem Cell Biobank, which is supported by the Cambridge NIHR Biomedical Research Centre, Wellcome Trust - MRC Stem Cell Institute and the Cambridge Experimental Cancer Medicine Centre, UK. Z.F. was supported by CSC-Cambridge Trust Scholarship. T.M. was supported by NIHR Starter Grant for Clinical Lecturers (SGL018/1032). D.F. was supported by Associazione Italiana Ricerca sul Cancro (AIRC-Fellowship 20930 for Abroad) and scholarships from Società Italiana di Ematologia (SIE) and Associazione "Amici di Beat Leukemia Dr. Alessandro Cevenini ONLUS" and AIL Bologna ODV. H.R.B. and J.H. were supported by the Medical Research Council (MC_UU_00015/2). S.M.-F was supported by the National Health Service Blood and Transplant (United Kingdom). S.M.-F was supported by European Union's Horizon 2020 research (ERC-2014-CoG-648765). S.M.-F was supported by MRC-AMED grant MR/V005421/1. S.M.-F was supported by a Programme Foundation Award (C61367/A26670) from Cancer Research UK. This research was funded in whole, or in part, by the Wellcome Trust 203151/Z/16/Z] and the UKRI Medical Research Council [MC_PC_17230]. For the purpose of open access, the author has applied a CC BY public copyright licence to any Author Accepted Manuscript version arising from this submission.

## Author contributions

Conceptualization: Z.F., T.M., J.H., E.J.B., A.L.G., C.N.H., and S.M.F. Trial coordination: S.Fox, R.S.F., and A.J. Methodology: J.E.M., L.G., M.P., Z.F., and R.H.B. Funding: C.N.H. and S.M.F., sponsors. Investigation: Z.F., G.C.F., T.M., D.F., J.R., R.G., H.R.B., E.H., A.R.M., A.J.M., S.K., J.E., N.M.B., M.J., S.Francis, F.J.C., J.C., M.F.M.M., F.W., S.N., D.M., M.W.D., M.S., and H.E. Visualization: Z.F., R.H.B., and S.M.F. Writing: Z.F., R.H.B., and S.M.F. Supervision: C.N.H. and S.M.F.

## Competing interests

C.N.H. reports funded research from Novartis; speaker fees from Novartis, Janssen, CTI, Celgene, and Medscape; and advisory board membership for Incyte, CTI, Sierra Oncology, Novartis, Celgene, Roche, AOP pharma, Geron, and AstraZeneca. A.J.M. reports funded research from Novartis, Celgene/BMS and consultancy for Abbvie, CTI, and Gilead. S.K. reports funded research and advisory board membership for Novartis. J.E. reports advisory board membership for Novartis, Incyte and Celgene/BMS. M.F.M. reports consultancies for Celgene and BMS; and advisory board membership for Novartis, Jazz, and Abbvie. S.N. reports speaker fees for Takeda, Celgene, Novartis, MSD and Alexion. D.M. reports consultancy for Incyte, Pfizer, Novartis, and Bristol-Myers Squibb. M.W.D. reports funded research from Blueprint Medicine Corporation and advisory board membership for Bristol-Myers Squibb, Novartis, Gilead, Pfizer, Jazz, Takeda, and Astellas. The remaining authors declare no competing financial interests. A.L.G. reports speaker fees from Novartis and advisory board membership AOP pharma and BMS.

## Additional information

Zijian Fang[1,2,3], Giuditta Corbizi Fattori[1,2,3,22], Thomas McKerrell[1,3,4,22], Rebecca H. Boucher[5], Aimee Jackson[5], Rachel S. Fletcher[5], Dorian Forte[1,2,3], Jose-Ezequiel Martin[6], Sonia Fox[5], James Roberts[2], Rachel Glover[2], Erica Harris[2], Hannah R. Bridges[7], Luigi Grassi[2], Alba Rodriguez-Meira[8], Adam J. Mead[8], Steven Knapper[9], Joanne Ewing[10], Nauman M. Butt[11], Manish Jain[12], Sebastian Francis[13], Fiona J. Clark[10], Jason Coppell[14], Mary F. McMullin[15], Frances Wadelin[16], Srinivasan Narayanan[17], Dragana Milojkovic[18], Mark W. Drummond[19], Mallika Sekhar[20], Hesham ElDaly[4], Judy Hirst[7], Maike Paramor[1], E. Joanna Baxter[2], Anna L. Godfrey[4], Claire N. Harrison[21] ✉ & Simón Méndez-Ferrer[1,2,3] ✉

[1]Wellcome-MRC Cambridge Stem Cell Institute, Cambridge, UK. [2]Department of Haematology, University of Cambridge, Cambridge, UK. [3]NHS Blood and Transplant, Cambridge, UK. [4]Cambridge University Hospitals NHS Foundation Trust, Cambridge, UK. [5]Cancer Research UK Clinical Trials Unit, University of Birmingham, Birmingham, UK. [6]Cancer Molecular Diagnostic Laboratory, Department of Oncology, University of Cambridge, Cambridge, UK. [7]MRC Mitochondrial Biology Unit, University of Cambridge, Cambridge, UK. [8]NIHR Biomedical Research Centre and MRC Molecular Haematology Unit, Weatherall Institute of Molecular Medicine, University of Oxford, Oxford, UK. [9]School of Medicine, Cardiff University, Cardiff, UK. [10]University Hospitals Birmingham NHS Foundation Trust, Birmingham, UK. [11]The Clatterbridge Cancer Centre NHS Foundation Trust, Liverpool, UK. [12]St James University Hospital, Leeds, UK. [13]Sheffield Teaching Hospitals NHS Trust, Sheffield, UK. [14]Royal Devon and Exeter Hospital, Exeter, UK. [15]Queens University, Belfast, UK. [16]Nottingham University Hospital, Nottingham, UK. [17]University Hospital Southampton NHSFT, Southampton, UK. [18]Imperial College Healthcare NHS Trust, London, UK. [19]Beatson West of Scotland Cancer Centre, Glasgow, UK. [20]University College Hospital London, London, UK. [21]Guy's and Saint Thomas' NHS Foundation Trust, London, UK. [22]These authors contributed equally: Giuditta Corbizi Fattori, Thomas McKerrell. ✉e-mail: Claire.Harrison@gstt.nhs.uk; sm2116@cam.ac.uk

