## [Peer Review File · Nature Communications]

Reviewers' Comments:

Reviewer #1:

Remarks to the Author:

Harrison as well as Mendez-Ferrer present here a very interesting combination of basic research and clinical trial. These are the most interesting kinds of papers targeting an audience in field of oncology.

While the basic research is really solid, the overall outcome of the trial is that Tamoxifen is not exhibiting sufficient activity. This is made difficult by an extensively heterogenous study population with different types of prior therapy and very different subtypes of MPN. This can be used to examine potential safety concerns but beyond this, a real statement regarding efficacy can not be given. The preclinical data indeed supported a clinical trials but considering that only 10% of all patients responded to therapy represents a rather very low response rate. This can always happen considering that preclinical data and clinical data can be different and does not deter the high quality of the trial.. However I would very much caution, that any Phase III trial would be recommended considering the very poor response. Besides this it is also very important to publish negative results and I would recommend publishing this ms.

Reviewer #2:

Remarks to the Author:

This is a very well designed and impactful work that methodically examines the molecular basis of the response to tamoxifen using response and clinical data and patient samples from a Phase-II, multicenter, single-arm clinical trial in patients with myeloproliferative neoplasms (MPNs). The authors show that among responders (in all 8, with 3 achieving primary outcome of more that 50% allele reduction and 5 with more that 25% allele reduction) 6 carry JAK2V617F-mutation while 2 of them carry CALRdel52 mutation. Subsequently, the study focuses on the mechanism of the response to JAK2V617F+ carriers. In those, the authors that tamoxifen acts by inducing apoptosis through suppressing activation of JAK2V617F+ induce STAT5 and oxidative phosphorylation genes and by inhibiting mitochondrial complex-I, activating proapoptotic integrated stress response (ISR) and decreasing pathogenic JAK2 signaling. The study involves analysis and cross-conformation of the results in human cell lines, patients cells and mouse model. The results are solid and the work sound. Some comments below aim to further delineate the basis of the selective sensitivity of JAK2V617F+ cells to tamoxifen and to understand the difference on the magnitude of the response.

Specifically:

1. How specific to JAK2V617F+ cells was the response to tamoxifen? It is not clear whether among the non-responders, there were any patients who carried the JAK2V617F+ mutation. If there were, are the tamoxifen modulated pathways on STA5 and ISR activation affected?
2. Did the authors examine whether cells with the CALRdel52 mutation are subject to the same effects and activation pathway following tamoxifen treatment?
3. Can the difference in the time and magnitude of response to tamoxifen in JAK2V617F+ patients be modified and explored in JAK2V617F+ cells of can be modelled in patient cells? For example does treatment with tamoxifen during serial relating assays affect the differentiation or the stemness of mutant cells or mutant allele contain in in cells from patients?
4. Have the authors compared the transcriptional profile of responding and non-responding JAK2V617F+ mutant cells and compared to CALRdel52 cells in response to tamoxifen?
5. Does the JAK2V617F mutation specifically sensitize cells to the effects of tamoxifen. Is the mitochondrial respiratory complex I to induce apoptosis specific to JAK2V617F-mutated cells? Has this pathway been examined cells without JAK2V617F mutation or with the CALRdel52 mutation.

Reviewer #3:

Remarks to the Author:

This article reports results of a Phase II single-arm clinical trials to evaluate potential efficacy and safety of tamoxifen in patients with stable myeloproliferative neoplasms (MPN). The trial enrolled 38 patients with 3 reported to have $\geq 50\%$ reduction in peripheral blood JAK2-V617F, CALR 5bp insertion (exon 9) or CALR 52bp deletion (exon 9) mutant allele burden at 24 weeks. The trial met the respecified success threshold to warrant further investigation of tamoxifen as therapy for MPN. The manuscript is well written. The study design is clean and straight forward. A few minor comments for the authors to consider.

1. Given this is a small study with several secondary and exploratory endpoints to evaluate activities of various molecular signature, there is a high likelihood of false positive conclusions. The study results should be interpreted with caution. The article title seems to over state the evidence obtained from the trial.

2. I would suggest clearly stating in the abstract and the conclusion that the results of this study warrants further investigation of tamoxifen as potential therapeutic for MPN.

Reviewer #4:

Remarks to the Author:

This manuscript describes a Phase-II, multicenter, single-arm clinical trial testing tamoxifen in patients with MPN. Based on pre-clinical data from the group that demonstrated JAK2V617F cells may be vulnerable to tamoxifen, the primary outcome was $\geq 50\%$ allele burden reduction at 24 weeks. This is a unique and forward-thinking endpoint in MPN clinical trials. Three out of 38 patients achieved a $\geq 50\%$ reduction and 5/38 additional patients showed $\geq 25\%$ reductions. An important finding in this paper was that the identification of a potential predictive signature in the baseline peripheral blood HSPCs of responders. This could potentially be clinically useful to identify those patients who are likely to respond and guide in treatment selection.

Critiques:

1. In supplemental data file S1 it would be helpful instead to list raw allele burdens. From my interpretation of the table the baseline column is a raw value, then the 12w and 24wk columns are % change from baseline. If this is the case then there are some numbers that are impossible. For example, TNO2 starts at a 100% VAF at baseline, then at week 12 is +17%. It would be impossible to have a VAF of 117%. TNO15 starts with a baseline VAF of 92% then it increases by 66% at wk 12 which would also be $> 100\%$.

2. One important item that is not included in the manuscript is additional mutations in this cohort (such as Tet2, Dnmt3a, ASXL1, IDH1,2....). For example, it is conceivable that a specific non-MPN driver mutation may be conserved among the responders.

3. Figure 2E shows ratio of mut/WT colonies after tamoxifen but this does not glean information about the CHANGE in % mutant with tamoxifen. What should be shown instead is the %WT/het/hom at baseline and at wk 24 for each of the patients, for example with a stacked bar. The % mutant colonies for the responders should decrease given the reduction in their bulk VAF. Also, showing WT/het/hom will also give potential information regarding whether homozygous cells are more sensitive than heterozygous JAK2VF cells.

4. In Figure 2F CD34+ from peripheral blood is used. Because this is a very rare population in the PB it would be important to show representative examples of gating schemes and document #'s and frequencies of CD34+ cells that were retrieved from each of the patients.

5. All mechanistic cell line work is done in JAK2V617F cells (which is reasonable), however 2/5 of the 25% VAF responders had CALR. It would bolster the manuscript to include a CALR cell line.

6. I am confused about Figure 7 and there are a number of discrepancies.

-Shouldn't EPO induce pSTAT5 in EPOR JAK2WT cells? But it doesn't in your figure.

-Why at baseline do the JAK2VF EPOR cells have less pSTAT5 than EPOR JAK2WT? Since JAK2V617F leads to constitutive activation of EPOR JAK2VF EPOR cells should have pSTAT5 even without stimulation.

-Why is % pSTAT5 plotted on Fig 7D rather than for example MFI?

-In Fig 7E which cells are shown - EPOR JAK2VF or EPOR JAK2VF?

Many thanks to all four reviewers for their constructive comments, which have greatly improved our manuscript. Over the past two months we have performed additional experiments and analyses that have strengthened the conclusions of our manuscript.

The main changes include:

- 1) Figure 7: we have performed new experiments and analyses and have revised gating strategy for STAT5 phosphorylation in BAF3 cells (Figure 7D). We have changed the description of these results in the manuscript (P.6 L.9-12): "Constitutive STAT5 activation, measured as phosphorylation at residue Y694, was detected in Ba/F3 cells expressing mutant JAK2^{V617F} without erythropoietin stimulation. STAT5 phosphorylation was increased in cytokine-stimulated Ba/F3 cells expressing both WT and mutant JAK2; this effect was significantly blocked by 4OH-TAM in the JAK2^{V617F+} cells (Fig. 7D-E)."
- 2) Figure 8: revised representative Western Blot for STAT5/JAK2 phosphorylation (highlight the constitutive phosphorylation of STAT5/JAK2 in HEL cells) (Figure 8E-G).
- 3) New Supplementary Figure 5 contains new experiments using a CALR-mutant cell line (MARIMO) which extends the effects of tamoxifen to CALR-mutant cells, explaining the effects observed in the clinical study. These new results have been described in the revised manuscript (P. 5, L.37-45): "Two MPN patients carrying CALR mutations showed a partial response to tamoxifen administration at 24w, indicative of the potential effect of tamoxifen on HSPCs carrying CALR mutations. To directly examine the effects of 4OH-TAM on CALR-mutated human cell metabolism and survival, we took advantage of the MARIMO cell line⁴⁰, which harbors a 61-basepair (bp) deletion in CALR exon 9, similar to the 52-bp deletion generated by the CALR^{del52} mutation⁴¹. Resembling the effects observed in human JAK2^{V617F+} cell lines, 4OH-TAM suppressed cellular respiration in MARIMO cells and reduced OXPHOS-derived ATP synthesis (fig. S5A-D). The supplementation of monomethyl succinate protected MARIMO cells from 4OH-TAM-induced cell death (fig. S5E), suggesting that tamoxifen induces apoptosis of cells carrying CALR mutations through a mechanism shared with JAK2^{V617F} mutation, which involves respiratory complex I inhibition."
- 4) We have updated the baseline VAF value in Supplementary Table 1 to reflect the comparator value used for allele burden change calculations and to avoid confusion.
- 5) We have acknowledged that "These results warrant further investigation of tamoxifen in MPN, with careful consideration of thrombotic risk" (Abstract, last sentence).
- 6) In the Discussion, we explain that "These results warrant further investigation of tamoxifen as potential therapeutic for MPN in larger studies, after careful consideration of the risk of thrombosis." (P.6, L.37-47). We have acknowledged that "the conclusions rely on a small patient cohort and warrant confirmation in future larger studies" (P.7, L.1-4).

All the changes in the manuscript and our responses to the reviewers' comments are marked with blue font.

Below is a point-by-point response to all of the reviewers' comments. We hope that the revised manuscript is considered to be acceptable for publication at *Nature Communications*.

Reviewer #1 – Metabolism, haematopoiesis, leukemia

Harrison as well as Mendez-Ferrer present here a very interesting combination of basic research and clinical trial. These are the most interesting kinds of papers targeting an audience in field of oncology. While the basic research is really solid, the overall outcome of the trial is that Tamoxifen is not exhibiting sufficient activity. This is made difficult by an extensively heterogeneous study population with different types of prior therapy and very different subtypes of MPN. This can be used to examine potential safety concerns but beyond this, a real statement regarding efficacy cannot be given. The preclinical data indeed supported a clinical trials but considering that only 10% of all patients responded to therapy represents a rather very low response rate. This can always happen considering that preclinical data and clinical data can be different and does not deter the high quality of the trial. However I would very much caution, that any Phase III trial would be recommend considering the very poor response. Besides this it is also very important to publish negative results and I would recommend publishing this ms.

Many thanks for considering the study very interesting, solid and for recommending publication. We have strengthened in the revised manuscript the need to expand these findings in larger studies designed to evaluate efficacy in a more homogenous patient population. We believe that such studies would be strongly guided by the results presented in the current manuscript.

Reviewer #2 – Haematopoiesis, MPN, sequencing

This is a very well designed and impactful work that methodically examines the molecular basis of the response to tamoxifen using response and clinical data and patient samples from a Phase-II, multicenter, single-arm clinical trial in patients with myeloproliferative neoplasms (MPNs). The authors show that among responders (in all 8, with 3 achieving primary outcome of more than 50% allele reduction and 5 with more than 25% allele reduction) 6 carry JAK2V617F-mutation while 2 of them carry CALRdel52 mutation. Subsequently, the study focuses on the mechanism of the response to JAK2V617F+ carriers. In those, the authors that tamoxifen acts by inducing apoptosis through suppressing activation of JAK2V617F+ induce STAT5 and oxidative phosphorylation genes and by inhibiting mitochondrial complex-I, activating proapoptotic integrated stress response (ISR) and decreasing pathogenic JAK2 signaling. The study involves analysis and cross-conformation of the results in human cell lines, patient cells and mouse model. The results are solid and the work sound. Some comments below aim to further delineate the basis of the selective sensitivity of JAK2V617F+ cells to tamoxifen and to understand the difference on the magnitude of the response.

Many thanks for the highly encouraging comments regarding the design, impact and robustness of the study, and for the very helpful suggestions to further delineate the selective sensitivity, which we address below.

Specifically:

1. How specific to JAK2^{V617F} cells was the response to tamoxifen? It is not clear whether among the non-responders, there were any patients who carried the JAK2^{V617F} mutation. If there were, are the tamoxifen modulated pathways on STAT5 and ISR activation affected?

The TAMARIN trial led to 3 major responses (allele burden reduction over 50% at 24w) and 3 partial responses (allele burden reduction between 25% and 50% at 24w) among 25 MPN patients carrying the JAK2^{V617F} mutation (**Figure R1**). This result suggests that a subset of JAK2^{V617F} MPN patients are particularly sensitive to tamoxifen treatment and prompted the laboratory studies described in the manuscript to determine the susceptibility features. Figure 7C-E shows a much higher sensitivity of BA/F3 cells expressing JAK2^{V617F} mutation, compared with those cells expressing wild-type JAK2. In-depth research using multiple human cell lines carrying the JAK2^{V617F} mutation (HEL, SET-2, UKE-1) confirmed their high sensitivity.

Figure R1. Oncogenic drivers in TAMARIN study subjects.

We have performed further studies to compare the sensitivity of human leukemic cell lines with/without JAK2^{V617F} mutation. In **Figure R2**, human leukemic cell lines carrying the JAK2^{V617F} mutation (HEL and SET2) show a higher susceptibility to 4-OH-tamoxifen, compared with human leukemic cell lines lacking JAK2^{V617F} (OCI-AML3 and MV-4-11). These results suggest that JAK2^{V617F} mutation sensitizes the cells to tamoxifen treatment (most likely due to its capacity to inhibit pathogenic JAK-STAT signalling, as shown in our study).

Figure R2. A higher tamoxifen sensitivity is detected in JAK2^{V617F}-mutated human leukemic cell lines (HEL and SET2), compared with non-mutated cell lines (OCI-AML3 and MV-4-11).

However, UKE-1 cells cultured under two different conditions showed different sensitivity due to metabolic signatures that also distinguished HSPCs from responders and non-responders in the TAMARIN study, despite the shared presence of JAK2^{V617F}. Therefore, we have further investigated this metabolic dependency by comparing the transcriptome of CD34⁺ HSPCs from TAMARIN responders and 4 control healthy individuals identically processed and sequenced in parallel, as internal control. Similar to the previous comparison of HSPCs from responders and non-responders, a higher enrichment of OXPHOS-related genes was found in responder HPSCs compared with HSPCs from healthy individuals. The signatures of apoptosis, STAT5-mediated transcription and unfolded protein response are only significant in the comparison of responders vs healthy individuals (**Figure R3**), but not in non-responders vs healthy controls (**Figure R4**), highlighting the unique molecular signature of tamoxifen-sensitive CD34⁺ HSPCs from responders.

Figure R3. GSEA of HSPCs from TAMARIN responders and control healthy individuals (Healthy_Con).

Figure R4. GSEA of HSPCs from TAMARIN non-responders and control healthy individuals (Healthy_Con).

2. Did the authors examine whether cells with the CALR^{del52} mutation are subject to the same effects and activation pathway following tamoxifen treatment?

10 MPN patients with CALR mutations (5 patients CALR 5bp INS, CALR^{ins5}, and 5 patients with CALR 52bp DEL, CALR^{del52}) were included in TAMARIN trial (Figure R1), and two of them showed a partial response to tamoxifen administration at 24w (33% allele burden reduction in one patient with CALR 5bp INS mutation, 25% allele burden reduction in one patient with CALR 52bp DEL). Although the limited number of patients in this Phase-II study does not allow to investigate associations with different driver mutations, the proportion of overall responders is similar across the 3 mutations. Therefore, the results suggest that a potential response is not limited to JAK2^{V617F}-mutated HSPCs, but might be found also in a proportion of CALR-mutated HSPCs. This is consistent with the fact that HSPCs carrying CALR mutations display a similar hyperactivation of JAK2/STAT5 signalling

(Araki *et al.*, 2016; Chachoua *et al.*, 2016; Elf *et al.*, 2016; Marty *et al.*, 2016) and increased level of unfolded protein response (Salati *et al.*, 2017; Ibarra *et al.*, 2022). Therefore, it is reasonable to speculate that tamoxifen may eliminate mutant HSPCs carrying CALR mutations in sensitive patients.

Regarding possible molecular differences in the response depending on the oncogenic driver, our RNAseq analysis was mainly focused on JAK2^{V617F}-mutated patients for homogeneity and comparability, and only one patient with CALR 5bp INS (a partial responder) could be subjected to the same analysis. With this limitation in mind, we have now explored the effect of tamoxifen on HSPCs carrying CALR mutation using gene set variation analysis (GSVA), which is a single-sample gene set enrichment method (Hänzelmann, Castelo and Guinney, 2013) that we have used to score the activation level of pathway using the RNAseq data of HSPCs from the only available responder carrying CALR mutation at baseline, 12w and 24w. Resembling the JAK2V617F-mutated cells lines (fig. S2A-C), CALR-mutant MPN HSPCs from the responder patient show downregulation of estrogen signalling-related genes upon tamoxifen treatment (**Figure R5**). Similar to the OXPHOS inhibition observed in JAK2^{V617F}-mutated HSPCs and cell lines upon tamoxifen treatment, HSPCs from CALR^{ins5}mutated responder suggest downregulation of genes related to oxidative phosphorylation, mitochondrial activity and cellular respiration (**Figure R5**). More importantly, STAT5 activity was also downregulated in post-treatment HSPCs from patient with CALR^{ins5} (**Figure R5**).

Figure R5. Transcriptomic analysis of HSPCs from CALR^{ins5}-mutated MPN responder. Gene set variation analysis at baseline, 12w or 24w after tamoxifen treatment reveals a downregulation of estrogen signalling, oxidative phosphorylation and STAT5 activity after tamoxifen treatment, resembling the molecular mechanism of tamoxifen on HSPCs carrying JAK2^{V617F} mutation. The activity scores of samples at 12w and 24w were normalized to the baseline score.

To directly examine the effects of 4-OH-tamoxifen on CALR-mutated human cell metabolism and survival, we took advantage of the MARIMO cell line, which carries a 61-basepair (bp) deletion in CALR exon 9 and produces a novel C terminus, similar to the 52-bp deletion generated by the CALR^{del52} mutation (Yoshida *et al.*, 1998; Kollmann *et al.*, 2015). Resembling the effects observed in human JAK2^{V617F+} cell lines, 4OH-tamoxifen suppressed cellular respiration in MARIMO cells, with diminished OXPHOS-derived ATP synthesis and reduced maximal and basal respiration (**Figure R6A-D**). Like in human JAK2^{V617F+} cells, supplementation of the respiratory complex II substrate monomethylsuccinate (MMS) protected MARIMO cells from 4OH-tamoxifen-induced apoptosis (**Figure R6E**), implying that tamoxifen induces apoptosis through respiratory complex one inhibition.

We have included the results from MARIMO cells in the new Supplementary Figure 5. These new results have been described in the manuscript (P. 5, L.37-45): "Two MPN patients carrying CALR mutations showed a partial response to tamoxifen administration at 24w, indicative of the potential effect of tamoxifen on HSPCs carrying CALR mutations. To directly examine the effects of 4OH-TAM on CALR-mutated human cell metabolism and survival, we took advantage of the MARIMO cell line⁴⁰, which harbors a 61-basepair (bp) deletion in CALR exon 9, similar to the 52-bp deletion generated by the CALR^{del52} mutation⁴¹. Resembling the effects observed in human JAK2^{V617F+} cell lines, 4OH-TAM suppressed cellular respiration in MARIMO cells and reduced OXPHOS-derived ATP synthesis (fig. S5A-D). The supplementation of monomethyl succinate protected MARIMO cells from 4OH-TAM-induced cell death (fig. S5E), suggesting that tamoxifen induces apoptosis of cells carrying CALR mutations through a mechanism shared with JAK2^{V617F} mutation, which involves respiratory complex I inhibition."

Figure R6. Tamoxifen inhibits mitochondrial respiration of MARIMO cells.

(A-D) 4OH-TAM suppressed the respiration of MARIMO cells, leading to reduced oxygen consumption rate (OCR) (A), basal respiration (B), maximal respiration (C) and OXPHOS-derived ATP generation (D).

(E) The activation of respiratory complex II by water soluble succinate (MMS, mono-methyl succinate) bypassed respiratory complex I inhibition and protected cell viability after 4OH-TAM treatment. These results indicate that tamoxifen inhibits respiratory complex I and induces apoptosis in CALR-mutated MARIMO cells similarly to JAK2V617F⁺ cells.

Although these results warrant validation in a larger series of patients, together with the results using MARIMO cells, they strongly suggest a similar mechanism of action of tamoxifen in HSPCs from MPN responders, regardless of the oncogenic driver mutation (JAK2/CALR).

3. Can the difference in the time and magnitude of response to tamoxifen in JAK2V617F⁺ patients be modified and explored in JAK2V617F⁺ cells of can be modelled in patient cells? For example does treatment with tamoxifen during serial relating assays affect the differentiation or the stemness of mutant cells or mutant allele contain in in cells from patients?

We have performed additional *in vitro* colony-forming unit (CFU) assays using peripheral blood mononuclear cells (PB MNCs) available from study subjects at 24 weeks to evaluate the effect of tamoxifen on the viability and differentiation of wild-type and mutant cells before and after 24-week tamoxifen treatment, which is different from the *ex vivo* 4OH-TAM treatment of baseline samples. CFU-C genotyping shows, after 24-week *in vivo* tamoxifen administration, a reduced balance of JAK2V617F⁺ colonies, compared with WT colonies, in PB MNCs samples from two partial responders, but not in three non-responders' samples (Figure R7), suggesting that a long treatment of tamoxifen selectively eliminates the mutant HSPCs, similar with *ex vivo* 4OH-TAM treatment of baseline samples.

Figure R7. Tamoxifen inhibits mitochondrial respiration of MARIMO cells.

HSPCs measured as colony-forming units in culture (CFU-Cs) from study patients' peripheral blood mononuclear cells treated. Reduced (**C**) or unchanged (**D**) HSPC numbers upon *ex vivo* 4OH-TAM treatment of baseline samples are consistent with the allele burden reductions observed in the same patients after 24w tamoxifen treatment (allele burden reductions after 24w are marked with orange $\geq 25\%$, $n = 2$; red $< 25\%$, $n = 3$). unpaired *t* test.

Interestingly, in one partial responder (TNO 35 with 25.47% reduction of mutant allele burden at 24W), a reduction of Burst-forming unit-erythroid (BFU-E) colony numbers is observed after 24h 4-OH-tamoxifen incubation, suggesting that 4-OH-tamoxifen treatment may interfere with the differentiation of erythroid precursor cell (which is overactivated in $JAK2^{V617F}$ -driven PV). A consistent but more pronounced effect was observed in PB MNCs from the same patient after 24w tamoxifen administration *in vivo*: a 3-fold reduction of BFU-E and CFU-G/M/GM suggests defective myeloid differentiation after long-term tamoxifen treatment in tamoxifen responder (**Figure R8A**). Of note, colony genotyping before and after tamoxifen treatment (both *in vitro* and *in vivo*) indicates a marked reduction of $JAK2^{V617F+}$ HSPCs and relative preservation or expansion of unmutated colonies (**Figure R8B**). Although these promising results would require confirmation in independent patients and samples which are unfortunately not available within the current study, they add to the more pronounced effects of tamoxifen in the mutant cells, compared with WT cells, already shown in our study.

Figure R8. CFU colony number (A) and genotyping results (B) of $JAK2^{V617F+}$ TAMARIN responder. **A, *In vitro* or *in vivo* treatment with tamoxifen reduces myeloid colonies, suggestive of impaired myeloid differentiation upon inhibition of pathogenic JAK-STAT signalling. **B**, *In vitro* or *in vivo* treatment with tamoxifen reduces the frequency of $JAK2^{V617F+}$ colonies (heterozygous, HET or homozygous, HOM), compared with unmutated (WT) colonies.**

4. Have the authors compared the transcriptional profile of responding and non-responding $JAK2^{V617F+}$ mutant cells and compared to CALRdel52 cells in response to tamoxifen?

As only one patient with CALR 5bp INS (a partial responder) was included in our RNAseq database, it is not sufficient to perform a comparable statistical analysis with other HSPCs carrying $JAK2^{V617F}$ mutation. However, a single-sample gene set enrichment method, gene set variation analysis (GSVA), was used to score the biological activity of HSPCs from this patient with CALR 5bp

INS at baseline, 12w and 24w (Hänzelmann, Castelo and Guinney, 2013). Resembling the results in JAK2V617F+ mutant cells from responders, the downregulation of oxidative phosphorylation and STAT5-related genes was observed in HSPCs with CALR 5bp INS mutation after tamoxifen treatment (**Figure R5** shown above).

5. Does the JAK2V617F mutation specifically sensitize cells to the effects of tamoxifen. Is the mitochondrial respiratory complex I to induce apoptosis specific to JAK2V617F-mutated cells? Has this pathway been examined cells without JAK2V617F mutation or with the CALRdel52 mutation.

We have performed additional experiments to evaluate tamoxifen sensitivity in human leukaemic cell lines without JAK2^{V617F} mutation (OCI-AML3 cells and MV-4-11 cells). 4-OH-tamoxifen displayed a higher toxicity on JAK2-mutated cells compared with those without JAK2^{V617F} mutation (**Figure R2**), which is consistent with our previous data using Ba/F3 cells expressing human wild-type and mutant JAK2 kinase in the manuscript (**Figure 7C-E**). This is due to the specific inhibition of pathological JAK2^{V617F}-STAT5 activation, which confers JAK2^{V617F+} cells a survival advantage and resistance to physiological apoptosis, compared with JAK2^{V617F-} cells.

As explained above, the MARIMO cell line was used to examine the proapoptotic mechanism of tamoxifen on JAK2^{V617F-} cells with the CALR^{del52} mutation. 4OH-TAM suppressed the cellular respiration of MARIMO cells and reduced OXPPOS-derived ATP synthesis (**Figure R6A-e**). The activation of respiratory complex II by succinate also rescued MARIMO cells from 4OH-TAM-induced apoptosis. This result is consistent with the fact that cells carrying CALR mutations display a similar hyperactivation of JAK2/STAT5 signalling (Araki *et al.*, 2016; Chachoua *et al.*, 2016; Elf *et al.*, 2016; Marty *et al.*, 2016) and increased level of unfolded protein response (Salati *et al.*, 2017; Ibarra *et al.*, 2022) and suggests that, that tamoxifen also induces apoptosis of CALR-mutated cells through respiratory complex I inhibition. In fact, CALR-mutant cells displayed a higher sensitivity to tamoxifen compared with JAK2^{V617F+} cells, which may be due to their increased integrated stress response/unfolded protein response. Unfolded protein response is molecular hallmark of tamoxifen responders and has been reported in HSPCs carrying CALR mutation (Salati *et al.*, 2017; Ibarra *et al.*, 2022). Moreover, a high expression of UPR-related genes (e.g. ATF3, ATF6, XBP1 and EIF2AK3(PERK)) is found in the tamoxifen-sensitive UKE-1 cell in deprivation of horse serum, resembling the high UPR/ISR signature in HSPCs from tamoxifen responders (**Figure R9**). We have found a synergistic effect of tamoxifen and UPR inducers (that increases intracellular UPR/ISR level) on MPN cell lines (**Figure R10**). Together, these results suggest that the elevation of UPR/ISR level confers sensitivity to tamoxifen. Thus, it is possible that the integrated stress response activation by tamoxifen triggers apoptosis of CALR-mutated HSPCs. This conclusion is supported by the two tamoxifen responders carrying CALR mutations in the TARMARIN trial, as well as the decreased cell viability and complex I activity in tamoxifen-treated MARIMO cells. However, these findings require further investigation and validation, which are not feasible due to the lack of sufficient study samples, and beyond the scope of the current study.

Figure R9. A higher expression of UPR-related genes is detected in tamoxifen-sensitive JAK2^{V617F+} cells cultured under serum deprivation.

Figure R10. UPR inducers sensitise MPN cells to tamoxifen. Tunicamycin (TM) pretreatment promotes tamoxifen-induced apoptosis in HEL cells. The synergistic effect of tamoxifen and thapsigargin (TG) overcomes tamoxifen resistance in UKE-1 cells.

Reviewer #3 - Biostatistics, clinical trials (Remarks to the Author): This article reports results of a Phase II single-arm clinical trials to evaluate potential efficacy and safety of tamoxifen in patients with stable myeloproliferative neoplasms (MPN). The trial enrolled 38 patients with 3 reported to have $\geq 50\%$ reduction in peripheral blood JAK2-V617F, CALR 5bp insertion (exon 9) or CALR 52bp deletion (exon 9) mutant allele burden at 24 weeks. The trial met the respecified success threshold to warrant further investigation of tamoxifen as therapy for MPN. The manuscript is well written. The study design is clean and straight forward. A few minor comments for the authors to consider.

1. Given this is a small study with several secondary and exploratory endpoints to evaluate activities of various molecular signature, there is a high likelihood of false positive conclusions. The study results should be interpreted with caution. The article title seems to over state the evidence obtained from the trial.

Thanks for this suggestion. We have added the following paragraph to the abstract to acknowledge that further investigation is warranted:

"These results demonstrate the safety and activity of tamoxifen inhibiting mitochondrial respiration, modulating ISR and reducing pathogenic JAK-STAT signalling and mutant allele burden in a subset of MPN patients who could potentially be prospectively identified based on their transcriptome at baseline. These results warrant further investigation of tamoxifen as potential therapeutic for MPN, with careful consideration of thrombotic risk."

We have emphasised in the discussion the limitation due to the study size.

2. I would suggest clearly stating in the abstract and the conclusion that the results of this study warrants further investigation of tamoxifen as potential therapeutic for MPN.

Thanks for this suggestion. We agree and have added this conclusion to the end of the abstract and the discussion.

Reviewer #4 - MPN clinical trials (Remarks to the Author): This manuscript describes a Phase-II, multicenter, single-arm clinical trial testing tamoxifen in patients with MPN. Based on pre-clinical data from the group that demonstrated JAK2V617F cells may be vulnerable to tamoxifen, the primary outcome was $\geq 50\%$ allele burden reduction at 24 weeks. This is a unique and forward-thinking endpoint in MPN clinical trials. Three out of 38 patients achieved a $\geq 50\%$ reduction and 5/38 additional patients showed $\geq 25\%$ reductions. An important finding in this paper was that the identification of a potential predictive signature in the baseline peripheral blood HSPCs of responders. This could potentially be clinically useful to identify those patients who are likely to respond and guide in treatment selection.

Critiques: 1. In supplemental data file S1 it would be helpful instead to list raw allele burdens. From my interpretation of the table the baseline column is a raw value, then the 12w and 24wk columns are % change from baseline. If this is the case then there are some numbers that are impossible. For example, TNO2 starts at a 100% VAF at baseline, then at week 12 is +17%. It would be impossible to have a VAF of 117%. TNO15 starts with a baseline VAF of 92% then it increases by 66% at wk 12 which would also be $> 100\%$.

Thank you for the comment and apologies for the confusion. The baseline column is a raw value, and the 12w and 24wk columns are the ratio of change from baseline to the screening raw value. Thus, the 12w and 24wk columns are not additive to the baseline column.

We have changed the column names as (%change ratio) in the updated supplementary table one.

The TAMARIN statistical analysis plan states that our comparator is the repeat screening measurement taken at the same time point e.g. week 12 and 24. In this case, even though the measurement of mutant allele burden from baseline sample had been done at the screening (**Original screening burden**, which was used for patient enrolment in the trial), the raw allele burden from the corresponding screening samples were measured again along with 12w or 24w samples (to account for the inherent technical variability and minimise its impact), which would be used to calculate the corresponding allele burden reduction at 12w or 24w as **Comparator**. For example, the baseline measurement of TNO2 is 100% at baseline, but the repeated measurement at 12w showed that the raw allele burdens are 84% and 99% at baseline and 12w respectively, so the **Comparator** value is 84% instead of 100%. In this case, 17% increase in mutant allele burden at 12w was obtained based on the calculation aforementioned. The variations in measurement do not affect the robustness of our conclusion because 1) primary outcome is the allele burden reduction at 24w, not affected by the variations at 12w, 2) the allele burden measurement at 24w have been repeated more than once in responder samples to make sure the observation of reduced allele burden is consistent throughout the different batches.

The Study Protocol established the use of the repeat value at 24w as the comparator. Only in the rare cases where the repeat screening was not available, the repeated screening at 12w or the original screening value (if the repeat was not available) was used.

Measurement time point	Baseline	12-week samples	24-week samples	Change Ratio in allele burden (%)
Screening (baseline)	A			
12-week	B (repeated measurement at 12w)	C		(C-B)/B*100
24-week	D (repeated measurement at 24w)	E (repeated measurement at 24w)	F	(F-D)/D*100

Table 1. The calculation of change in mutant allele burden

2. One important item that is not included in the manuscript is additional mutations in this cohort (such as Tet2, Dnmt3a, ASXL1, IDH1,2....). For example, it is conceivable that a specific non-MPN driver mutation may be conserved among the responders.

The reviewer is absolutely correct that non-driver somatic mutations are important in MPN pathogenesis. However, whether these mutations can affect tamoxifen sensitivity cannot be determined due to the large number of myeloid hotspot mutations (the panel normally used at Cambridge University Hospital includes 50) and the small size of this pilot study. Considering the relatively low frequency of these 50 mutations and the number of responders and non-responders in the TAMARIN study, it would be impossible to establish significant associations.

In this study, we focused on JAK2 mutation as tamoxifen selectively targets the JAK2^{V617F+} HSPCs in preclinical mouse model (Sánchez-Aguilera *et al.*, 2014). Single-cell RNAseq of mouse HSPCs population also revealed that tamoxifen administration results in the selective depletion of haematologic population with high JAK-STAT activity (Kucinski *et al.*, 2022). These indicate the selectivity of tamoxifen on JAK2^{V617F} HSPCs + in human MPNs. Indeed, the activation of JAK2-STAT pathway is positively associated with mutant allele burden reduction in MPN patients after tamoxifen treatment. Mechanistically, we demonstrated that tamoxifen inhibits the pathogenic activation of JAK2^{V617F}-STAT axis through metabolic reprogramming. However, it remains unclear if the non-driver mutational profile directly impacts JAK-STAT signalling in mutant HSPCs carrying driver mutations on *JAK2* or *CALR*. The biological functions of additional co-occurring somatic mutations are less relevant to cytokine signalling, but rather affect epigenetic regulators (DNMT3A, TET2, IDH1/2), chromatin modifiers (ASXL1, EZH2), RNA splicing (U2AF1, SF3B1, SRSF2, ZRSR2), DNA repair (TP53) and transcriptional factors (RUNX1 and NFE2) (Klampfl *et al.*, 2013; Grinfeld *et al.*, 2018). Thus, it is challenging to correlate the acquisition of non-MPN driver mutations with the primary and secondary outcome of TAMARIN study (i.e. the elimination of mutant clones with driver mutations). In this case, tamoxifen may affect the sub-clones with non-MPN driver mutations independent of JAK-STAT signalling, but further validation on *in vitro* and *in vivo* models will be required to delineate whether the effect of tamoxifen on clonal hierarchies contribute to the elimination of malignant clones with driver mutations, which are beyond the scope of the current study.

3. Figure 2E shows ratio of mut/WT colonies after tamoxifen but this does not glean information about the CHANGE in % mutant with tamoxifen. What should be shown instead is the %WT/het/hom at baseline and at wk 24 for each of the patients, for example with a stacked bar. The % mutant colonies for the responders should decrease given the reduction in their bulk VAF. Also, showing WT/het/hom will also give potential information regarding whether homozygous cells are more sensitive than heterozygous JAK2VF cells.

Thank you for this suggestion. The stacked bar plots of the colony genotype from patient samples were shown in **Figure R10** and **Figure R11**. Indeed, consistent with the reduction of bulk mutant allele burden, tamoxifen treatment (both 24w tamoxifen treatment (see **Figure R7**) or *in vitro* 24H 4OH-TAM incubation (see Fig 2E) resulted in the expansion of WT colonies and elimination of mutant colonies. However, it is challenging to conclude that a different sensitivity towards tamoxifen was found between colonies with homozygous and heterogenous mutation. In fact, the profile of JAK2^{V617F} mutation was highly variable among patients sample. For instance, while approximately 95% of colonies from TNO 32 and TNO 13 samples was wild-type at baseline, most of colonies from TNO 15 samples were JAK2^{V617F}-positive, with a moderate reduction after 24H 4OH-TAM incubation. Despite of a drop of mutant clones in responder sample after tamoxifen treatment, the effect of tamoxifen on mutant HSPCs was not obvious in the stacked bar in some patient samples. Thus, to consistently present the present the selectivity of tamoxifen on mutant HSPCs, the ratio of mut/WT colonies was used in the manuscript.

Figure R11. Stacked bar plots showing WT/mutant clones from responder HSPCs with or without *in vitro* 4OH-TAM incubation.

Figure R12. Stacked bar plots showing WT/mutant clones from responder HSPCs at baseline and 24w.

4. In Figure 2F CD34⁺ from peripheral blood is used. Because this is a very rare population in the PB it would be important to show representative examples of gating schemes and document #'s and frequencies of CD34⁺ cells that were retrieved from each of the patients.

Thank you for your question. We performed CD34⁺ enrichment of patient peripheral blood samples using CD34 MicroBead Kit UltraPure (Cat Number: 130-100-453, Miltenyi Biotec) and obtain a high purity (90%) of CD34⁺ HSPCs (**Figure R13**), which were then sorted for RNAseq preparation.

Figure R13. A representative example of gating strategy for CD34⁺ HSPCs after immunomagnetical isolation of peripheral blood samples.

5. All mechanistic cell line work is done in JAK2V617F cells (which is reasonable), however 2/5 of the 25% VAF responders had CALR. It would bolster the manuscript to include a CALR cell line.

Thank you for the suggestion. To directly examine the effects of 4-OH-tamoxifen on CALR-mutated human cell metabolism and survival, we took advantage of the MARIMO cell line, which carries a 61-basepair (bp) deletion in CALR exon 9 and produces a novel C terminus, similar to the 52-bp deletion generated by the CALR^{del52} mutation (Yoshida *et al.*, 1998; Kollmann *et al.*, 2015). Resembling the effects observed in human JAK2^{V617F+} cell lines, 4OH-tamoxifen suppressed cellular respiration in MARIMO cells, with diminished OXPHOS-derived ATP synthesis and reduced respiration coupling efficiency (**Figure R6A-C**). Like in human JAK2^{V617F+} cells, supplementation of the respiratory complex II substrate monomethylsuccinate (MMS) protected MARIMO cells from 4OH-tamoxifen-induced apoptosis (**Figure R6D above**), implying that tamoxifen induces apoptosis through respiratory complex one inhibition.

6. I am confused about Figure 7 and there are a number of discrepancies.

- Shouldn't EPO induce pSTAT5 in EPOR JAK2WT cells? But it doesn't in your figure.
- Why at baseline do the JAK2VF EPOR cells have less pSTAT5 than EPOR JAK2WT? Since JAK2V617F leads to constitutive activation of EPOR JAK2VF EPOR cells should have pSTAT5 even without stimulation.
- Why is % pSTAT5 plotted on Fig 7D rather than for example MFI?
- In Fig 7E which cells are shown - EPOR JAK2VF or EPOR JAK2WT?

Thank you very much for pointing this out. The discrepancies were due to one experimental repeat with extremely low pSTAT5 measurement in JAK2WT and JAK2V617F cells. To confirm this, we have performed additional experiments and have reanalysed the data using a consistent gating strategy (**Figure R11A**). The raw data of the three independent experiments is shown in **Figure R12**. As the reviewer expected, EPO increases the percentage of pSTAT5-positive cells in both JAK2WT and JAK2V617F EPOR cells. Without EPO stimulation, a higher percentage of pSTAT5-positive cells was found in JAK2V617F EPOR cells, compared with WT JAK2V617F EPOR cells.

The Figure 7D in manuscript has been replaced with an updated figure. We have changed the description of these results in the manuscript (P.6 L.9-12): "Constitutive STAT5 activation, measured as phosphorylation at residue Y694, was detected in Ba/F3 cells expressing mutant JAK2^{V617F} without erythropoietin stimulation. STAT5 phosphorylation was increased in cytokine-stimulated Ba/F3 cells expressing both WT and mutant JAK2; this effect was significantly blocked by 4OH-TAM in the JAK2^{V617F+} cells (Fig. 7D-E)."

The reason we used % pSTAT5 is because STAT5 staining sometimes showed a double peak or merging peak in BA/F3 cells expressing WTJAK2 (**Figure R12**), with a pSTAT5-positive population and a pSTAT5-negative population. Of note, a single peak was always shown in JAK2VF EPOR BAF3 cells after EPO stimulation, supporting the high cytokine sensitivity in JAK2VF cells. The gating of pSTAT5-positive cell was based on the isotype control, consistent through all samples in three independent

experiments, to quantify the pSTAT5-positive population. The quantification using geometric mean is available in **Figure R11B**, which shows a similar result as % pSTAT5.

In figure 7E, BA/F3 cells express both EPOR and JAK2VF.

Figure R14. Updated quantification of STAT5 phosphorylation in BA/F3 cells by % pSTAT5 (A) and Geometric mean (B).

-TPO
Vehicle

+TPO
Vehicle

+TPO
4OH-TAM

Gate	Number	%Total	%Gated	X-Med
All	9,926	99.10	100.00	4.45
S	5,295	52.87	53.34	7.39
Gate X-GMean				
All	4.54			
S	7.77			

Gate	Number	%Total	%Gated	X-Med
All	9,957	99.48	100.00	3.94
S	4,616	46.12	46.36	6.53
Gate X-GMean				
All	4.04			
S	7.09			

Gate	Number	%Total	%Gated	X-Med
All	6,806	98.51	100.00	7.03
S	4,612	66.75	67.76	9.05
Gate X-GMean				
All	5.47			
S	9.27			

Gate	Number	%Total	%Gated	X-Med
All	9,761	96.50	100.00	5.83
S	7,167	70.86	73.42	7.03
Gate X-GMean				
All	5.87			
S	7.57			

Gate	Number	%Total	%Gated	X-Med
All	9,810	97.37	100.00	14.89
S	9,545	94.74	97.30	15.05
Gate X-GMean				
All	14.43			
S	15.19			

Gate	Number	%Total	%Gated	X-Med
All	9,787	96.11	100.00	3.87
S	4,489	44.08	45.87	6.26
Gate X-GMean				
All	3.68			
S	6.65			

Gate	Number	%Total	%Gated	X-Med
All	8,501	82.46	100.00	0.48
S	4	0.04	0.05	4.94
Gate X-GMean				
All	0.47			
S	4.86			

Isotype control

-TPO
Vehicle

+TPO
Vehicle

+TPO
4OH-TAM

Gate	Number	%Total	%Gated	X-Med
All	9,849	98.17	100.00	2.96
S	2,107	21.00	21.39	5.17
Gate X-GMean				
All	2.95			
S	5.51			

Gate	Number	%Total	%Gated	X-Med
All	9,940	98.89	100.00	5.69
S	7,514	74.75	75.59	6.48
Gate X-GMean				
All	5.46			
S	6.77			

Gate	Number	%Total	%Gated	X-Med
All	5,527	96.78	100.00	1.75
S	1,259	22.05	22.78	7.21
Gate X-GMean				
All	2.06			
S	7.49			

Gate	Number	%Total	%Gated	X-Med
All	9,919	99.04	100.00	4.69
S	6,084	60.75	61.34	5.78
Gate X-GMean				
All	4.61			
S	6.13			

Gate	Number	%Total	%Gated	X-Med
All	9,834	97.55	100.00	5.62
S	7,676	76.14	78.06	6.23
Gate X-GMean				
All	5.57			
S	6.59			

Gate	Number	%Total	%Gated	X-Med
All	9,787	96.44	100.00	3.62
S	3,992	39.34	40.79	5.75
Gate X-GMean				
All	3.30			
S	6.10			

Gate	Number	%Total	%Gated	X-Med
All	8,592	82.85	100.00	0.51
S	11	0.11	0.13	6.16
Gate X-GMean				
All	0.49			
S	7.77			

Isotype control

BAF3
Wild Type JAK2

BAF3
JAK2V617F

BAF3
Wild Type JAK2

BAF3
JAK2V617F

Figure R15. Raw data of experiments in Figure R11.

References

- Araki, M., Yang, Y., Masubuchi, N., Hironaka, Y., Takei, H., Morishita, S., Mizukami, Y., Kan, S., Shirane, S., Edahiro, Y., Sunami, Y., Ohsaka, A. and Komatsu, N. (2016) 'Activation of the thrombopoietin receptor by mutant calreticulin in CALR-mutant myeloproliferative neoplasms', *Blood*, 127(10), pp. 1307–1316. doi:10.1182/blood-2015-09-671172.
- Bose, P., Nazha, A., Komrokji, R.S., Patel, K.P., Pierce, S.A., Al-Ali, N., Sochacki, A., Shaver, A., Ma, W., Su, X., Daver, N.G., DiNardo, C.D., Garcia-Manero, G., Loghavi, S., Bueso-Ramos, C., Kantarjian, H.M., Sekeres, M.A., Savona, M.R., Maciejewski, J.P. and Verstovsek, S. (2018) 'Mutational landscape of myelodysplastic/myeloproliferative neoplasm-unclassifiable', *Blood*, 132(19), pp. 2100–2103. doi:10.1182/blood-2018-05-848473.
- Chachoua, I., Pecquet, C., El-Khoury, M., Nivarthi, H., Albu, R.-I., Marty, C., Gryshkova, V., Defour, J.-P., Vertenoel, G., Ngo, A., Koay, A., Raslova, H., Courtoy, P.J., Choong, M.L., Plo, I., Vainchenker, W., Kralovics, R. and Constantinescu, S.N. (2016) 'Thrombopoietin receptor activation by myeloproliferative neoplasm associated calreticulin mutants', *Blood*, 127(10), pp. 1325–1335. doi:10.1182/blood-2015-11-681932.
- De-Morgan, A., Meggendorfer, M., Haferlach, C. and Shlush, L. (2021) 'Male predominance in AML is associated with specific preleukemic mutations', *Leukemia*, 35(3), pp. 867–870. doi:10.1038/s41375-020-0935-5.
- Elf, S., Abdelfattah, N.S., Chen, E., Perales-Patón, J., Rosen, E.A., Ko, A., Peisker, F., Florescu, N., Giannini, S., Wolach, O., Morgan, E.A., Tothova, Z., Losman, J.A., Schneider, R.K., Al-Shahrour, F. and Mullally, A. (2016) 'Mutant calreticulin requires both its mutant C-terminus and the thrombopoietin receptor for oncogenic transformation', *Cancer Discovery*, 6(4), pp. 368–381. doi:10.1158/2159-8290.CD-15-1434.
- Grinfeld, J., Nangalia, J., Baxter, E.J., Wedge, D.C., Angelopoulos, N., Cantrill, R., Godfrey, A.L., Papaemmanuil, E., Gundem, G., MacLean, C., Cook, J., O'Neil, L., O'Meara, S., Teague, J.W., Butler, A.P., Massie, C.E., Williams, N., Nice, F.L., Andersen, C.L., Hasselbalch, H.C., Guglielmelli, P., McMullin, M.F., Vannucchi, A.M., Harrison, C.N., Gerstung, M., Green, A.R. and Campbell, P.J. (2018) 'Classification and Personalized Prognosis in Myeloproliferative Neoplasms', *New England Journal of Medicine*, 379(15), pp. 1416–1430. doi:10.1056/nejmoa1716614.
- Hänzelmann, S., Castelo, R. and Guinney, J. (2013) 'GSVA: Gene set variation analysis for microarray and RNA-Seq data', *BMC Bioinformatics*, 14. doi:10.1186/1471-2105-14-7.
- Ibarra, J., Elbanna, Y.A., Kurylowicz, K., Ciboddo, M., Greenbaum, H.S., Arellano, N.S., Rodriguez, D., Evers, M., Bock-Hughes, A., Liu, C., Smith, Q., Lutze, J., Baumeister, J., Kalmer, M., Olschok, K., Nicholson, B., Silva, D., Maxwell, L., Dowgielewicz, J., Rumi, E., Pietra, D., Casetti, I.C., Catricala, S., Koschmieder, S., Gurbuxani, S., Schneider, R.K., Oakes, S.A. and Elf, S.E. (2022) 'Type I but Not Type II Calreticulin Mutations Activate the IRE1α/XBP1 Pathway of the Unfolded Protein Response to Drive Myeloproliferative Neoplasms', *Blood Cancer Discovery*, 3(4), pp. 298–315. doi:10.1158/2643-3230.bcd-21-0144.
- Kamphuis, P., Van Zeventer, I.A., De Graaf, A.O., Salzbrunn, J.B., Van Bergen, M.G.J.M., Dinmohamed, A.G., Van Der Reijden, B.A., Schuringa, J.J., Jansen, J.H. and Huls, G. (2023) 'Sex Differences in the Spectrum of Clonal

Hematopoiesis', *HemaSphere*, 7(2), p. E832. doi:10.1097/HS9.0000000000000832.

Klampfl, T., Gisslinger, H., Harutyunyan, A.S., Nivarthi, H., Rumi, E., Milosevic, J.D., Them, N.C.C., Berg, T., Gisslinger, B., Pietra, D., Chen, D., Vladimer, G.I., Bagienski, K., Milanesi, C., Casetti, I.C., Sant'Antonio, E., Ferretti, V., Elena, C., Schischlik, F., Cleary, C., Six, M., Schalling, M., Schönegger, A., Bock, C., Malcovati, L., Pascutto, C., Superti-Furga, G., Cazzola, M. and Kralovics, R. (2013) 'Somatic mutations of calreticulin in myeloproliferative neoplasms', *New England Journal of Medicine*, 369(25), pp. 2379–2390. doi:10.1056/NEJMoa1311347.

Kollmann, K., Nangalia, J., Warsch, W., Quentmeier, H., Bench, A., Boyd, E., Scott, M., Drexler, H.G. and Green, A.R. (2015) 'MARIMO cells harbor a CALR mutation but are not dependent on JAK2/STAT5 signaling', *Leukemia*, 29(2), pp. 494–497. doi:10.1038/leu.2014.285.

Kucinski, I., Campos, J., Barile, M., Severi, F., Bohin, N., Moreira, P.N., Allen, L., Lawson, H., Haltali, M.L.R., Kinston, S.J., O'Carroll, D., Kranc, K.R. and Göttgens, B. (2022) 'A time and single-cell resolved model of hematopoiesis', *bioRxiv*, p. 2022.09.07.506735. Available at: <https://www.biorxiv.org/content/10.1101/2022.09.07.506735v1%0Ahttps://www.biorxiv.org/content/10.1101/2022.09.07.506735v1.abstract>.

Marty, C., Pecquet, C., Nivarthi, H., El-Khoury, M., Chachoua, I., Tulliez, M., Villeval, J.L., Raslova, H., Kralovics, R., Constantinescu, S.N., Plo, I. and Vainchenker, W. (2016) 'Calreticulin mutants in mice induce an MPL-dependent thrombocytosis with frequent progression to myelofibrosis', *Blood*, 127(10), pp. 1317–1324. doi:10.1182/blood-2015-11-679571.

Salati, S., Prudente, Z., Genovese, E., Pennucci, V., Rontautoli, S., Bartalucci, N., Mannarelli, C., Ruberti, S., Zini, R., Rossi, C., Bianchi, E., Guglielmelli, P., Tagliafico, E., Vannucchi, A.M. and Manfredini, R. (2017) 'Calreticulin Affects Hematopoietic Stem/Progenitor Cell Fate by Impacting Erythroid and Megakaryocytic Differentiation', *Stem Cells and Development*, 27(4), pp. 225–236. doi:10.1089/scd.2017.0137.

Sánchez-Aguilera, A., Arranz, L., Martín-Pérez, D., García-García, A., Stavropoulou, V., Kubovcakova, L., Isern, J., Martín-Salamanca, S., Langa, X., Skoda, R.C., Schwaller, J. and Méndez-Ferrer, S. (2014) 'Estrogen signaling selectively induces apoptosis of hematopoietic progenitors and myeloid neoplasms without harming steady-state hematopoiesis', *Cell Stem Cell*, 15(6), pp. 791–804. doi:10.1016/j.stem.2014.11.002.

Yoshida, H., Kondo, M., Ichihashi, T., Hashimoto, N., Inazawa, J., Ohno, R. and Naoe, T. (1998) 'A novel myeloid cell line, Marimo, derived from therapy-related acute myeloid leukemia during treatment of essential thrombocythemia: Consistent chromosomal abnormalities and temporary C-MYC gene amplification', *Cancer Genetics and Cytogenetics*, 100(1), pp. 21–24. doi:10.1016/S0165-4608(97)00017-4.

Reviewers' Comments:

Reviewer #2:

Remarks to the Author:

The authors have addressed extensively my concerns

Reviewer #3:

Remarks to the Author:

The authors had satisfactorily responded to my comments.

Reviewer #4:

Remarks to the Author:

Thank you for the thorough responses to the critiques. I have no further critiques to add.

Many thanks to all four reviewers for their constructive comments over the revision process, which have greatly improved our manuscript.

REVIEWERS' COMMENTS

Reviewer #2 (Remarks to the Author):

The authors have addressed extensively my concerns

Reviewer #3 (Remarks to the Author):

The authors had satisfactorily responded to my comments.

Reviewer #4 (Remarks to the Author):

Thank you for the thorough responses to the critiques. I have no further critiques to add.